# Incremental Extractive Opinion Summarization Using Cover Trees

**Somnath Basu Roy Chowdhury**[1], **Nicholas Monath**[2], **Kumar Avinava Dubey**[3],
**Manzil Zaheer**[2], **Andrew McCallum**[2], **Amr Ahmed**[3], **Snigdha Chaturvedi**[1]
[1] *UNC Chapel Hill*, [2] *Google DeepMind*, [3] *Google Research*
*{somnath, snigdha}@cs.unc.edu*
*{nmonath, avinavadubey, manzilzaheer, mccallum, amra}@google.com*

**Reviewed on OpenReview:** *https://openreview.net/forum?id=IzmLJ1t49R*

## Abstract

Extractive opinion summarization involves automatically producing a summary of text about an entity (e.g., a product's reviews) by extracting representative sentences that capture prevalent opinions in the review set. Typically, in online marketplaces user reviews accumulate over time, and opinion summaries need to be updated periodically to provide customers with up-to-date information. In this work, we study the task of extractive opinion summarization in an incremental setting, where the underlying review set evolves over time. Many of the state-of-the-art extractive opinion summarization approaches are centrality-based, such as CentroidRank (Radev et al., 2004; Chowdhury et al., 2022). CentroidRank performs extractive summarization by selecting a subset of review sentences closest to the centroid in the representation space as the summary. However, these methods are not capable of operating efficiently in an incremental setting, where reviews arrive one at a time. In this paper, we present an efficient algorithm for accurately computing the CentroidRank summaries in an incremental setting. Our approach, `CoverSumm`, relies on indexing review representations in a cover tree and maintaining a reservoir of candidate summary review sentences. `CoverSumm`'s efficacy is supported by a theoretical and empirical analysis of running time. Empirically, on a diverse collection of data (both real and synthetically created to illustrate scaling considerations), we demonstrate that `CoverSumm` is up to 36x faster than baseline methods, and capable of adapting to nuanced changes in data distribution. We also conduct human evaluations of the generated summaries and find that `CoverSumm` is capable of producing informative summaries consistent with the underlying review set.

## 1 Introduction

Opinion summarization (Hu & Liu, 2004; Medhat et al., 2014; Pang, 2008) is the process of automatically generating summaries of user reviews about entities (such as e-commerce products). The summarized text is useful in many ways, including assisting customers in making informed purchasing decisions and aiding sellers in understanding user feedback. Incremental opinion summarization is particularly important as online marketplaces continue to grow and the efficiency and accurate evolution of product summaries is paramount. Most of the current opinion summarization systems operate in a static setup – generating a summary based on the complete set of reviews. Updating opinion summaries with such static systems is expensive, as the entire set of reviews must be processed each time a new review arrives. This necessitates the development of techniques that can efficiently update summaries with changes in the review set. Even when utilizing the LLMs, such as GPT-4 (Bubeck et al., 2023), performing opinion summarization still poses challenges due to constraints like token limit as well as high computational cost in incremental settings (Bhaskar et al., 2023).

---

Code available here: https://github.com/brcsomnath/CoverSumm.

Opinion summarization systems can produce either abstractive (Isonuma et al., 2021; Kim Amplayo et al., 2022) or extractive summaries (Kim et al., 2011). While abstractive summaries allow for novel phrasing, they often suffer from hallucination (Ji et al., 2022), lack of faithfulness (Maynez et al., 2020) and interpretability (Saha et al., 2022), especially for larger review sets. Extractive summaries, which are the focus of this work, reduce concerns about hallucination and lack of faithfulness (Zhang et al., 2023) by producing a summary consisting only of a carefully selected subset of review sentences. Most extractive opinion summarization systems operate in an unsupervised setup due to the large volume of reviews in real-world settings. In the incremental setup, providing supervision becomes even more challenging, as it necessitates having oracle or human-written summaries at each time step, which is expensive. Therefore, we focus on unsupervised approaches to perform incremental extractive opinion summarization.

Unsupervised extractive summarization approaches assign saliency scores to review sentences and extract the ones with the highest scores as the summary (Peyrard, 2019; Angelidis et al., 2021). The quality and efficiency of producing the summary depend on the extractive summarization method. While previous work presented complex graph-based objectives that use lexical features to extract summarizing sentences (Erkan & Radev, 2004; Mihalcea & Tarau, 2004; Nenkova & Vanderwende, 2005), recent work has demonstrated that a simple centrality-based approach with learned representations for the input reviews produces state-of-the-art results across a wide variety of datasets (Chowdhury et al., 2023; 2022; Chu & Liu, 2019). This objective, known as CentroidRank (Radev et al., 2004), functions by calculating the centroid of review sentences in the representation space, and selecting the nearest neighbors of the centroid as the summary. Note that CentroidRank's objective is closely related to classic problems in machine learning such as finding the central nodes in a graph (Okamoto et al., 2008) or medoid(s) within a dataset (Bagaria et al., 2018; Baharav & Tse, 2019). Previous work (Chowdhury et al., 2022; 2023) has advanced the state-of-the-art in extractive summarization using CentroidRank approaches by improving the representation learning method for the input reviews. There is less work focusing on the algorithmic aspects of these methods. In particular, there is limited work in adapting these methods to incremental settings, where reviews are added over time. and a summary needs to be kept up-to-date with each addition.

In this paper, we focus on the task of incremental extractive opinion summarization, which involves the extraction of salient sentences from a continuous stream of reviews as they arrive. We propose a novel algorithm `CoverSumm`, that executes on centroid-based extractive summarization (Chowdhury et al., 2022; Li et al., 2023; Gholipour Ghalandari, 2017; Radev et al., 2004; Rossiello et al., 2017) in an incremental setup. Centroid-based incremental extractive summarization requires computing the $k$-nearest neighbours of the centroid (where $k$ is the number of sentences in the summary) at each point in time. `CoverSumm` efficiently performs incremental summarization by maintaining a small reservoir of input samples (reviews) without processing the entire review set at every time step. Empirical evaluation shows that `CoverSumm` is up to 36x faster than baselines. The speedup occurs as `CoverSumm` limits most of the nearest neighbour search queries to the reservoir instead of the entire review set. We perform experiments on large real-world review sets and show that generated summaries align with the aggregate user reviews. Our primary contributions are:

- We study the problem of extractive opinion summarization in an incremental setup, where a system generates an updated summary with each incoming user review (Section 2).
- We extend the paradigm of centroid-based summarization to an incremental setup, and propose `CoverSumm` that performs extractive summarization using cover trees (Section 3).
- We perform theoretical analyses to show that `CoverSumm` generates exact nearest neighbours (NN) and provide bounds for the number of NN queries, as well as maximum storage required (Section 3.1).
- We evaluate `CoverSumm` to show that it is significantly faster than baselines (up to 36x), and requires minimal additional space. We also perform experiments to gauge the quality of the generated summaries, and if the content of the summaries aligns with the aggregate user reviews (Section 4).

## 2 Preliminaries & Background

In this section, we first describe centroid-based extractive summarization in the incremental setting. Then, we present the data structures used for efficient nearest neighbor and range search in `CoverSumm`. Following that, we will outline some simple baselines that utilize these data structures.

## 2.1 Problem Formulation

In this section, we formally describe the problem of extractive summarization and then its extension in an incremental setting. Given a set of review sentences for a product, the objective of the extractive summarization system is to select a subset of sentences as the summary. Similar to prior works, we compute the saliency score for each review sentence and select a subset of sentences with high salience scores as the summary. We build on the paradigm of centroid-based extractive summarization techniques (Radev et al., 2004; Rossiello et al., 2017), where the saliency score of a review sentence is quantified as its distance to the centroid in the representation space. We assume access to a representation model that yields a numerical representation for input texts. In this paradigm of summarization, the system greedily selects $k$-nearest neighbours of the centroid as the output summary. Mathematically, given a set of $n$ sentence representations in $D$-dimensional space, $\mathcal{X} = [x_1, x_2, \ldots] \in \mathbb{R}^{n \times D}$, the summary representation with budget $k$ is $\mathbf{S} \in \mathbb{R}^{k \times D}$ and is computed as:

$$\mathbf{S} = \texttt{knn}(\mathcal{X}, \mu) \in \mathbb{R}^{k \times D}, \ \mu = \frac{1}{n} \sum_i x_i \in \mathbb{R}^D. \tag{1}$$

The final summary is a concatenation of review sentences whose representations are present in $\mathbf{S}$.

**Incremental Summarization Task**. We study this paradigm of summarization in an incremental setup, where at every time step $t$, a new representation $x_t$ arrives and we have the representation set $\mathcal{X}_t = [x_j]_{j=1}^t$. Using this set, the system should generate a summary $\mathbf{S}_t$ consistent with the formulation in Equation 1. We aim to develop efficient algorithms that accurately estimate $\mathbf{S}_t$. In this work, we primarily focus on reducing computation overhead, not storage space, as storing text representations is relatively inexpensive.

## 2.2 Efficient Nearest Neighbour Data Structures

Efficient retrieval of nearest neighbors of a centroid can be performed using index-based data structures. In this work, we will focus on cover tree-based index structures (Beygelzimer et al., 2006; Zaheer et al., 2019).

**Cover Trees** (Beygelzimer et al., 2006). Each node in a cover tree is associated with a representative point $x \in \mathcal{X}$. The nodes of the tree are arranged into a series of levels. Nodes in level $\ell$, denoted $C_\ell$, have an associated parameter $\gamma^\ell$ used to define the the following set of invariants:

- Covering: For each representation $x \in C_\ell$, there exists at least a representation $y \in C_{\ell-1}$ such that distance $d(x, y) \leq \gamma^\ell$. One such representation $y$ that satisfies the condition must be a parent of $x$.
- Separation: Any pair of representations $x, y \in C_\ell$ are separated by at least $d(x, y) > \gamma^\ell$.
- Nesting: If $x$ appears at level $\ell$, it must appear at all lower levels. Therefore, $C_\ell \subset C_{\ell-1}$.

Cover trees support efficient nearest neighbor search using the algorithms presented by Beygelzimer et al. (2006) and modified for use in our method (Algorithm 2). The construction operations (e.g., insert) of the cover tree can be performed in an incremental manner. For a detailed discussion of the running time of search and construction of cover trees, we refer readers to (Elkin & Kurlin, 2022; 2023).

**Stable Greedy (SG) Trees** (Zaheer et al., 2019). SG trees are defined similarly to cover trees, with a modified separation property. Rather than requiring all nodes in the level to be separated, only sibling nodes in the tree structure are required to be separated, e.g., all pairs of siblings $x, y \in C_\ell$ are at least $d(x, y) > \gamma^\ell$ apart. An illustration of an SG Tree is shown in Figure 1 (bottom left). Zaheer et al. (2019) demonstrated that SG trees empirically improve on the online construction time compared to cover trees. In the following sections, we will observe that in our proposed algorithm, construction poses more of a bottleneck than search queries. Therefore, in practice, we will use SG trees for our experiments. In presenting our proposed methods, we will refer to the tree structures as cover trees (since the results are general to the broader class of cover tree data structures of which SG trees are an instance).

## 2.3 Baselines

In this section, we will discuss baseline approaches for performing incremental centroid-based summarization before introducing our proposed algorithm, `CoverSumm`.

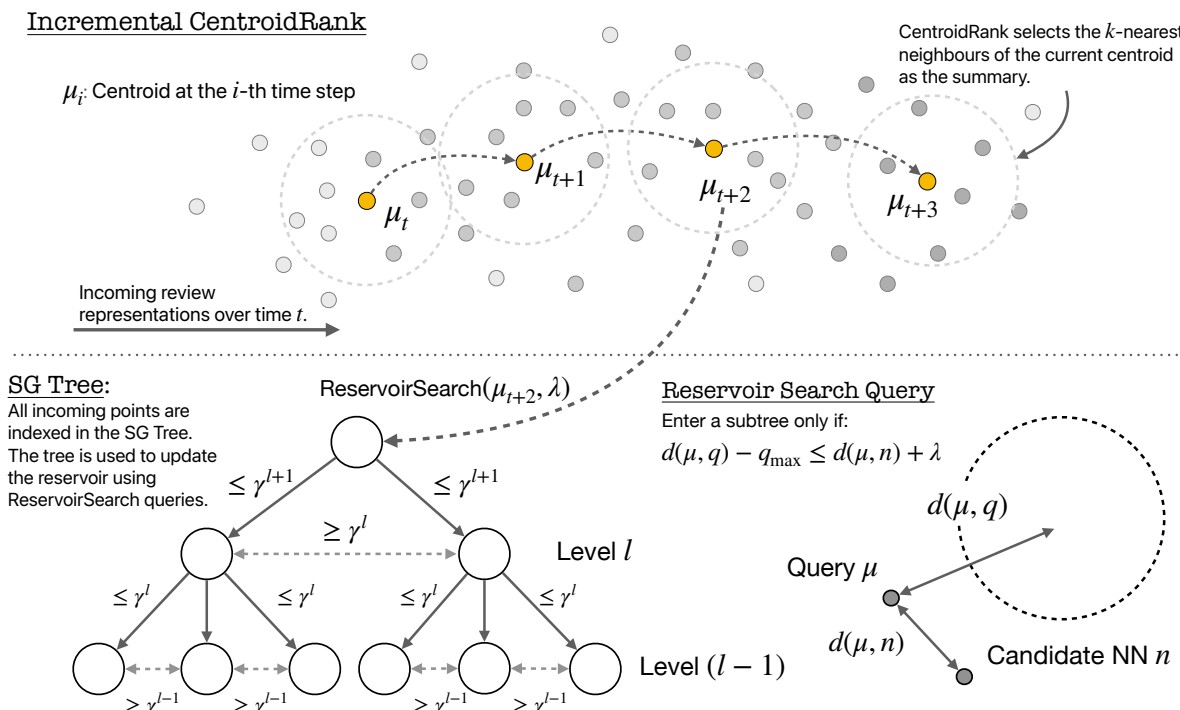

Figure 1: Overview of the incremental CentroidRank-based summarization task and the utility of nearest neighbour (NN) data structures. (Top): We show the CentroidRank task in the incremental setting, where the centroid ($\mu_i$) evolves over time. Additionally, we illustrate the benefits of maintaining a reservoir of candidate samples for NN retrieval. (Bottom): We show how utilizing efficient NN retrieval data structures, such as the SG Tree, can enhance the efficiency of incremental summarization using reservoir search (discussed in Section 3), particularly in scenarios where the reservoir does not have all the NNs the centroid.

**Brute Force Algorithm**. In this approach, we compute the centroid representation at every time step $\mu_t = 1/t \sum_{i=1}^{t} x_i \in \mathbb{R}^D$. Then, we compute the summary at step $t$: $\mathbf{S}_t = \texttt{knn}(\mathcal{X}_t, \mu_t)$. Brute-force computation of nearest neighbours (Equation 1) requires $O(t)$ time at each summarization step. Assuming a total of $n$ text representations in the input stream, the time complexity is $O(n^2)$.

**Naive Algorithm with Cover Tree**. The bulk of the computation in the brute force algorithm relies on $\texttt{knn}$ queries (as centroid is updated in $O(1)$ w.r.t. the number of instances). The running time of the $k$-nearest neighbor operation can be reduced by using any exact or approximate nearest neighbour retrieval data structures. We use cover trees (Beygelzimer et al., 2006) to improve the performance of the brute-force algorithm. At each time step $t$, we execute the following queries on a cover tree $\texttt{insert}(x_t)$ and $\texttt{knn}(\mu_t)$. For a total of $n$ reviews, this yields an overall time complexity of $O(n \log n)$. While our work uses cover tree because of its theoretical properties illustrated above, the index could be replaced with any efficient nearest neighbour retrieval index (e.g., $k$d-tree (Bentley, 1975), quad tree (Samet, 1984), r-tree (Guttman, 1984), HNSW (Malkov & Yashunin, 2018b), etc.) that supports similar kinds of incremental operations like insertions, nearest neighbour, and range search.

## 3 Cover Tree based Summarization (`CoverSumm`)

In this section, we present an efficient algorithm, `CoverSumm`, for performing centroid-based extractive summarization in an incremental setup. `CoverSumm` builds on the observation that as the number of reviews increases, there are smaller perturbations in their centroid, $\mu_t$ (assuming that the representations belong to the same distribution with a fixed centroid), thereby resulting in infrequent changes to the $k$-nearest neighbours of $\mu_t$. Motivated by this observation, we improve upon the Naive CT algorithm (Section 2.3) by

reducing the number of $\texttt{knn}(\mu_t)$ queries executed. This is achieved by maintaining a small reservoir $\mathcal{R}$ of representations that we anticipate to be the $k$-nearest neighbours of future centroids $\mu_{>t}$. This allows us to work with a small number of representations in $\mathcal{R}$ without having to execute expensive nearest neighbour queries on the entire set, $\mathcal{X}$. To describe our algorithm precisely, we first define the following propositions:

**Proposition 1** (Bounding distance to centroid). *Let $x_1, x_2, \ldots, x_t$ be i.i.d. samples from a distribution with centroid $\mu$ supported on $[-b/2, b/2]^D$ and $\mu_t = \sum_{i=1}^{t} x_i/t$. Then, the Euclidean distance of any sample $x$ from $\mu_t$ can be bounded with probability $(1-\delta)^D$ as:*

$$d(\mu_t, x) \leq d(\mu, x) + \sqrt{\frac{Db^2 \log(2/\delta)}{2t}}, \ \forall x \in [-b/2, b/2]^D. \tag{2}$$

We obtain the above result by applying the Hoeffding-Azuma inequality (Hoeffding, 1963) (more details in Appendix A.1) to each dimension of the vector followed by triangle inequality (proof in Appendix A.2). Next, we use the result from Proposition 1 to derive a bound on how much subsequent centroids shift over time. The detailed proof of Proposition 2 is provided in Appendix A.3.

**Proposition 2** (Bounding distance between subsequent centroids). *Let $x_1, x_2, \ldots, x_t$ be i.i.d. samples from a distribution with centroid $\mu$ supported on $[-b/2, b/2]^D$ and $\mu_t = \sum_{i=i}^{t} x_i/t$. Then, we can bound the Euclidean distance between subsequent centroids $\mu_t$ and $\mu_{t+i}$ with probability $(1-\delta)^{2D}$:*

$$d(\mu_t, \mu_{t+i}) \leq \sqrt{\frac{2Db^2 \log(2/\delta)}{t}}, \ \forall t \in [n-1], i \in [n-t]. \tag{3}$$

**Outline**. Equipped with these results, we present our summarization algorithm, $\texttt{CoverSumm}$ (Algorithm 1). The main idea behind our approach is to maintain a small reservoir of sentences $\mathcal{R}$ (with a maximum capacity of $c_{\max}$) along with the cover tree index (ct). The reservoir's capacity $\mathcal{R}$ is set to be more than the summary budget $k$. $\mathcal{R}$ facilitates the computation of $k$-nearest neighbours of $\mu_{t+i}$ $(i > 0)$, in the subsequent time steps instead of executing expensive knn queries on the entire cover tree. We initialize $\mathcal{R}$ with points close to the current centroid $\mu_t$ from the cover tree, ct. Naively applying a range search query in the cover tree can return all representations within a radius of $r$ to the input query $\mu_t$. To have more than $k$ representations in $\mathcal{R}$, the range search radius $r$, should be at least $r \geq d_k$ (where $d_k$ is the distance of the $k$-th nearest neighbour from $\mu_t$). We use Proposition 2 (which provides an estimate of how far the centroid may shift) to set the radius of the range search query as shown below:

$$r = d_k + \underbrace{\sqrt{2\alpha Db^2 \log(2/\delta)/t}}_{\lambda}, \tag{4}$$

where $\alpha$ is a hyperparameter. We set the confidence $\delta = O(1/t)$. The size of the reservoir $\mathcal{R}$ is proportional to the radius $r$, as more samples would be selected with a higher $r$. Depending on the reservoir budget $c_{\max}$, the radius can be tuned using the parameter, $\alpha$.

Therefore, re-initializing the reservoir requires the following queries to the cover tree: (a) $k$-nearest neighbour query to estimate $d_k$ (required in computing $r$ in Eqn. 4) and (b) range search query with the radius $r$. These queries are executed for the same point $\mu_t$ and their sequential execution involves quite a bit of redundant computation. A naive way to reduce this computation is through memorization. Instead, we present an elegant algorithm, *reservoir search*, to retrieve the reservoir elements given a threshold $\lambda$ and summary budget $k$ in Algorithm 2. This algorithm can retrieve the reservoir in a single tree traversal in $O(\log n)$.

**Reservoir Search**. In Algorithm 2, we use a modified version of the nearest neighbour search algorithm in SG Trees to populate the reservoir $\mathcal{R}$ whenever the distance to a node is within the range search radius (Line 13). This algorithm does not miss out on any potential reservoir candidates because at any time the $d_k$ estimate is more than the exact distance to the $k$-th neighbour. We perform a filtering step at the end to remove points outside the actual range search radius (Line 21). We also use the range radius to ignore nodes that do not have any children within the search radius (Line 15). We use the reservoir search method to

| **Algorithm 1** CoverSumm Algorithm | **Algorithm 2** CoverTree Reservoir Search |
|---|---|
| 1: **function CoverSumm**(Sentence $x_t$) | 1: **function ReservoirSearch**(Query $\mu_t$, Threshold $\lambda$, Summary Budget $k$): |
| 2:     Hyperparameters $\alpha$ and $c_{\max}$. | 2:     Candidate NN List $\mathcal{N} = \{\}$, Reservoir $\mathcal{R} = \{\}$ |
| 3:     $\mathcal{X}_t = \mathcal{X}_{t-1} \cup x_t$ | 3:     Queue $Q = \{\texttt{ct.root}()\}$ |
| 4:     $\mu_t = \mathbb{E}[\mathcal{X}_t]$ // updated in $O(1)$ | 4:     **while** $Q$ is not empty **do** |
| 5:     // insert $x_t$ in the cover tree | 5:         $q = Q$.pop() |
| 6:     $\texttt{ct.insert}(x_t)$ | 6:         // $\mathcal{N}$.max: max distance of any neighbour from $\mu_t$. At start $\mathcal{N}$.max $= \infty$. |
| 7:     // drift of $\mu_t$ from the last query $\mu_{\text{last}}$ | 7:         **if** $d(\mu_t, q) \leq \mathcal{N}$.max **then** |
| 8:     $\Delta = \|\mu_t - \mu_{\text{last}}\|$ | 8:             Insert $(q, d(\mu_t, q))$ into Sorted List $\mathcal{N}$ |
| 9:     **if** $\Delta \geq \lambda/2 \ \vee \ |\mathcal{R}| \geq c_{\max}$ **then** | 9:             **if** $\mathcal{N}$.size() $> k$ **then** $\mathcal{N}$.pop() |
| 10:         $\delta = O(1/t)$ // set confidence | 10:             $d_k = \mathcal{N}$.max // update $d_k$ |
| 11:         // compute threshold acc. to Eqn. 4 | 11:         **end if** |
| 12:         $\lambda = \sqrt{2\alpha D b^2 \log(2/\delta)/t}$ | 12:         $r = d_k + \lambda$ // set search radius |
| 13:         // reinitialize reservoir from cover tree | 13:         **if** $d(\mu_t, q) \leq r$ **then** $\mathcal{R} = \mathcal{R} \cup q$ |
| 14:         $\mathcal{R} = \texttt{ct.ReservoirSearch}(\mu_t, \lambda, k)$ | 14:         // don't explore nodes outside search range |
| 15:         // compute the radius of $\mathcal{R}$ | 15:         **if** $d(\mu_t, q) - q$.max $> r$ **then** continue |
| 16:         $r = \max\{d(\mu_t, q)|q \in \mathcal{R}\}$ | 16:         **for** each child $c$ of $q$ **do** |
| 17:         $\mu_{\text{last}} = \mu_t$ // update $\mu_{\text{last}}$ | 17:             **if** $d(\mu_t, c) - c$.max $\leq r$ **then** $Q = Q \cup c$ |
| 18:     **else** | 18:         **end for** |
| 19:         // add $x_t$ to $\mathcal{R}$ if it is within radius $r$ | 19:     **end while** |
| 20:         **if** $\|\mu_{\text{last}} - x_t\| \leq r$ **then** $\mathcal{R} = \mathcal{R} \cup x_t$ | 20:     // filter out points outside of range |
| 21:     **end if** | 21:     $\mathcal{R} = \{r \in \mathcal{R}|d(\mu_t, r) \leq \mathcal{N}$.max $+ \lambda\}$ |
| 22:     // form summary using kNN search in $\mathcal{R}$ | 22:     **return** $\mathcal{R}$ |
| 23:     $\mathbf{S}_t = \texttt{knn}(\mathcal{R}, \mu_t)$ | 23: **end function** |
| 24:     **return** $\mathbf{S}_t$ | |
| 25: **end function** | |

efficiently update the reservoir, $\mathcal{R}$, during incremental summarization in Algorithm 1. Next, we describe the outline of our incremental summarization algorithm, CoverSumm.

**CoverSumm Algorithm**. In Algorithm 1, at every time step $t$, we update the current centroid $\mu_t$ (Line 4) and insert $x_t$ into the cover tree ct (Line 6). We also compute the drift $\Delta$, which measures how much $\mu_t$ has shifted from $\mu_{\text{last}}$ (Line 8), when the last time reservoir search was performed. If the drift $\Delta$ was more than the threshold $\lambda/2$ or the reservoir is at capacity, it implies that the current reservoir $\mathcal{R}$ does not contain all the $k$-nearest neighbours of $\mu_t$ (proof in Proposition 4). In this case, we reinitialize the reservoir by executing reservoir search query on the cover tree (Line 14). If $\Delta$ is within the threshold, we update the $\mathcal{R}$ with current point $x_t$ if needed (Line 20). Finally, we compute the summary $\mathbf{S}_t$ by performing $k$-nearest neighbour search in the reservoir. The review text corresponding to the $k$-nearest neighbours is the output extractive summary. An illustration of the overall summarization algorithm is shown in Figure 1.

We can improve the efficiency of CoverSumm even further, by delaying the cover tree insertions (Line 6) to only when the drift exceeds the threshold (Line 9) as the cover tree is not utilized in the other computation flow. CoverSumm can also facilitate the deletion of reviews in cases where certain reviews need to be deleted due to spam or offensive content (more details in Appendix D). In the following section, we present several theoretical properties of CoverSumm.

### 3.1 Theoretical Analysis

In this section, we analyze properties of the efficiency and accuracy of CoverSumm, including its ability to produce the correct nearest neighbors for CentroidRank, the number of *reservoir search* queries required, the size of the reservoir used, and the rate of change of search queries.

**Proposition 3** (Correctness of Reservoir Search). *If ct is a SG Tree, then **ReservoirSearch**$(\mu, \lambda, k)$ returns all the neighbours of $\mu$ within a distance of $(d_k + \lambda)$, where $d_k$ is the distance of $\mu$ to its $k$-th nearest neighbour.*

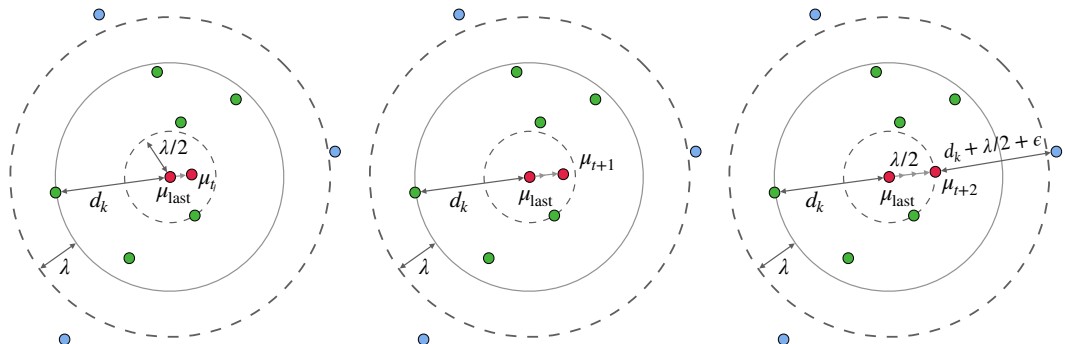

Figure 2: An illustration of `CoverSumm`'s operation over three consecutive time steps. Red circles ● represent centroids at different times, green circles ● indicate current nearest neighbors in reservoir $\mathcal{R}$, and blue circles ● denote representations outside $\mathcal{R}$. The figure displays the last query $\mu_{\text{last}}$ with radius $(d_{\text{k}} + \lambda)$. In the first two cases, the current centroid is within distance $\lambda$ of $\mu_{\text{last}}$, and the summary can be retrieved from the reservoir. The rightmost figure shows a boundary case where a representation is just outside the reservoir's boundary and summary computation requires querying the cover tree.

*Proof.* First, we note that $\mathcal{N}.\max = \infty$ at the beginning and $d_k$ (Line 10) is always more than or equal to the exact distance to the $k$-th NN during the algorithm. This means that Line 17 cannot throw away any grandparent of a nearest neighbour within distance $r = d_k + \lambda$. Similarly, Line 13 cannot ignore a nearest neighbour within distance $r$. Since, the reservoir $\mathcal{R}$ now contains all the possible NN candidates, Line 21 returns the set of exact nearest neighbours within distance $r$ of query $\mu$. □

**Proposition 4** (Exact nearest neighbours). *The $k$-nearest neighbours returned by Algorithm 1, $\mathbf{S}_t$ are the exact $k$-nearest neighbours of $\mu_t$, the mean at every time step $t$, i.e., $\mathbf{S}_t = knn(\mathcal{X}, \mu_t)$.*

*Proof.* We first note that $\mathcal{R}$ is initialized using reservoir search with radius $r = d_k + \lambda$, where $d_k$ is the distance of $\mu_t$ to its $k$-th neighbour. Therefore, for any radius $r \geq d_k$, the reservoir size is always $|\mathcal{R}| \geq k$.

Next, we consider the case when the updated centroid $\mu_t$ is within distance $\|\mu_t - \mu_{\text{last}}\| < \lambda/2$. In this case, the distance from $\mu_t$ of any representation not in the reservoir (blue circles ● in Figure 2 (left & center)) is at least $d(x, \mu_t) > d_{\min} = d_k + \lambda/2$, $\forall x \in \mathcal{X}_t \setminus \mathcal{R}$. Now, note that all points in $\mathbf{S}_{\text{last}} \in \mathcal{R}$, as the $\mathcal{R}$ has not been updated after that (also $|\mathbf{S}_{\text{last}}| = k$). Next, we can show that:

$$\forall x \in \mathbf{S}_{\text{last}}, \; d(x, \mu_t) \leq d(x, \mu_{\text{last}}) + d(\mu_t, \mu_{\text{last}})$$
$$< d_k + \lambda/2 = d_{\min} \tag{5}$$

This shows that the reservoir has at least $k$ points within $d_{\min}$ of $\mu_t$. Therefore, performing `knn` within $\mathcal{R}$ is exact. When the updated mean $\mu_t$ drifts more $\Delta \geq \lambda/2$ (shown in Figure 2 (right)), we perform a reservoir search query on the entire cover tree which is also exact. □

Now that we know that `CoverSumm` generates the exact nearest neighbours at all time steps. We focus on estimating the number of reservoir search queries (Line 14 in Algorithm 1) required to achieve it in the following proposition (proof in Appendix A.5).

**Proposition 5** (Number of reservoir search queries). *Assuming incoming points $x_t \in [-b/2, b/2]^D$, the number of reservoir search queries on the cover tree executed $n_{\text{rs}} = O(D \log n)$, where $n$ is the total number of points (e.g., review sentences) and $D$ is the dimensionality of the data.*

We observe that the reservoir search query count grows with the logarithm of the total number of representations $n$, unlike baseline approaches that execute nearest neighbour queries at every step. Although the overall complexity of our algorithm is $O(n \log n)$, since we perform insertions ($O(\log n)$ complexity) at each step, the constants involved for nearest neighbour and range search queries on cover trees are significantly larger. As a result, limiting the number of nearest neighbour queries leads to substantial speed improvements in practice.

Next, we investigate the storage cost involved in maintaining the reservoir $\mathcal{R}$ (detailed proof in Appendix A.6).

**Proposition 6** (Maximum reservoir size)**.** *For points arriving from a metric space $(\mathcal{X}, d)$, the upper bound of the reservoir size ($|\mathcal{R}|$) at large time steps ($t \gg 0$) is $O(k)$.*

The above result shows that the storage cost of the reservoir at large time steps is relatively low and does not scale with the number of points or dimensions of the data. We provide additional theoretical results about the interval between consecutive nearest neighbour cover queries in Appendix A.7.

## 4 Evaluation

In this section, we describe the dataset and baseline approaches used in our experiments.

**Datasets**. We evaluate `CoverSumm` on both synthetic and real-world datasets. For synthetic data, we experiment with two different setups: (a) we sample representations $x$ from a uniform distribution, where each representation serves as a proxy text representation, and (b) we sample synthetic data from an LDA process (more details in Appendix C) to mimic data from the textual domain. We use 10K vectors of $x \in \mathbb{R}^{100}$ for all experiments using synthetic data. We also perform experiments on two real-world datasets: (a) SPACE dataset (Angelidis et al., 2021) has $\geq$ 3K hotel reviews per entity with a total of $\geq$ 50K review sentences. We use sentence representations, $x \in \mathbb{R}^{8192}$, from the state-of-the-art extractive summarization system SemAE (Chowdhury et al., 2022) on SPACE. (b) Amazon US reviews (He & McAuley, 2016) have product reviews along with their temporal order. We only consider products with more than 1000 reviews in this setup. To simulate an incremental setup, we present an algorithm with one review sentence at a time and evaluate its performance by reporting the computation time and quality of the generated summaries.

**Baselines.** We compare the efficiency gains of `CoverSumm` with the following centroid-based baselines:

- Brute force: computes the $k$NN from the updated mean $\mu_t$ at every step.
- Naive CT: performs nearest neighbour query on the cover tree at each step.
- Naive HNSW: performs NN query on an HNSW (Malkov & Yashunin, 2018a) index at each step.
- Naive FAISS: performs NN query on a FAISS (Johnson et al., 2019) index at each step.
- `CoverSumm` (knn+range): uses a modification of the proposed algorithm (Algorithm 1), where a separate knn and range search query is executed on the cover tree instead of the reservoir search.
- `CoverSumm` (reservoir): implements the proposed online summarization approach in Algorithm 1.
- `CoverSumm` (lazy reservoir): modifies the `CoverSumm`'s summarization routine by delaying the insertions into the cover tree only when required (Line 9 is satisfied). This is the most efficient version of our proposed algorithm, and we will often refer to it as simply `CoverSumm`.

The above baselines can produce the exact nearest neighbours at every time step. We also compare `CoverSumm` with some naive variants that approximate the nearest neighbors to assess the trade-offs between nearest neighbor accuracy and runtime. The variants are listed below:

- `CoverSumm` (random): randomly decides whether an incoming point should be included in the reservoir $\mathcal{R}$.
- `CoverSumm` (decay $\lambda$): uses an exponential decay term as the threshold (independent of the current points) to decide whether a new point should be part of the reservoir. Specifically, we change Line 6 of the CoverSumm algorithm to $\Delta \leq c_1 \exp^{-c_2 T}$, where $c_1, c_2$ are hyperparameters.

We also compare the quality of `CoverSumm`'s summaries with different extractive summarization algorithms: SumBasic (Vanderwende et al., 2007), LexRank (Erkan & Radev, 2004), TextRank (Mihalcea & Tarau, 2004), Centroid-OPT (Ghalandari, 2017), and Latent Semantic Analysis (LSA) (Ozsoy et al., 2011). Centroid OPT is the most similar to CentroidRank. CentroidOPT constructs the summary by greedily selecting sentences that maximize the similarity of the constructed summary with the centroid. While CentroidRank measures the similarity between a sentence and the centroid, CentroidOPT does so for the summary. In our experiments, we use the SG Tree implementation available in `graphgrove` library (more details in Appendix E.1). We set hyperparameters by using grid search on a held-out development set for each dataset.

### 4.1 Main Results

We evaluate `CoverSumm` on several opinion summarization datasets and perform additional experiments to analyze its performance.

| | Algorithm | Uniform | | LDA | | SPACE | | AMAZON | |
|---|---|---|---|---|---|---|---|---|---|
| | | Time (s) | Acc. (%) | Time (s) | Acc. (%) | Time (s) | Acc. (%) | Time (s) | Acc. (%) |
| Exact | Brute force | 28.23 | 100.0 | 26.09 | 100.0 | 7.78 | 100.0 | 49.82 | 100.0 |
| | Naive CT | 26.37 | 100.0 | 25.21 | 100.0 | 3.23 | 100.0 | 17.53 | 99.6 |
| | Naive Faiss | 2.22 | 100.0 | 2.25 | 100.0 | 1.30 | 100.0 | 4.81 | 99.6 |
| | CoverSumm (knn+range) | 1.08 | 100.0 | 1.34 | 100.0 | 1.43 | 100.0 | 2.30 | 99.7 |
| | CoverSumm (reservoir) | 0.91 | 100.0 | 0.96 | 100.0 | 1.25 | 100.0 | 1.47 | 99.7 |
| | CoverSumm (lazy reservoir) | **0.87** | 100.0 | **0.92** | 100.0 | **1.14** | 100.0 | **1.36** | 99.7 |
| Appx. | Naive HNSW | 5.35 | 4.9 | 4.44 | 26.6 | 3.25 | 72.3 | 7.07 | 81.2 |
| | CoverSumm (random) | 0.53 | 0.2 | 0.59 | 0.2 | 1.02 | 2.3 | 0.62 | 0.5 |
| | CoverSumm (decay $\lambda$) | 1.28 | 1.0 | 0.70 | 8.9 | 1.35 | 10.7 | 0.53 | 2.3 |

Table 1: Total incremental summarization runtime (per entity) and nearest neighbour (NN) accuracy for different centroid-based algorithms for representations from synthetic (Uniform & LDA) and opinion review (SPACE & AMAZON) datasets. We observe that CoverSumm outperforms baseline approaches, showcasing up to 36x speedup. We also observe that reservoir search achieves up to 28% speedup compared to sequentially performing knn and range queries on the tree. Among the algorithms that retrieve *exact* NNs, we highlight the best time achieved in **bold**.

In Table 1, we compare the computation efficiency of CoverSumm with other centroid-based baseline methods in the incremental setting. Since centroid-based summarization relies on $k$NN retrieval, we evaluate the accuracy that the generated nearest neighbours exactly match the actual nearest neighbours. We report the average time required to complete the incremental summarization process per entity across 5 runs. For the synthetic dataset, we consider all 10K presentations belonging to a single entity. We consider an output to be incorrect if either any of the elements or their order does not match. We retrieve $k = 20$ nearest neighbours in our experiments. This serves as a proxy for the quality of the generated summary. We observe that CoverSumm outperforms baseline approaches by a significant margin, showcasing up to 36x speed gains while generating exact nearest neighbours. Moreover, we observe that CoverSumm (lazy reservoir) achieves the best time performance. This is expected as the implementation of performing insertions in a batch can be parallelized using SG Trees. In Figure 3, we show an illustration of how CoverSumm's runtime fares with baselines during the online summarization process on a synthetic dataset. In Table 1, we also observe that the speed gains on SPACE dataset are lower than others. This is due to the higher data dimension ($\mathbb{R}^{8192}$), as the confidence bound in Proposition 2 weakens with an increase in data dimension $D$, leading to more $k$NN queries. We analyze this phenomenon further in Section 4.2.

**Speedup.** We compare CoverSumm's runtime with other paradigms of extractive summarization in Table 4 on samples from the SPACE and AMAZON datasets.[1] These algorithms are quite expensive and some of them ran out of time for synthetic benchmarks (with >10K samples) reported in Table 1. A few algorithms like TextRank require access to text inputs and cannot be applied to synthetic data. We observe that our proposed algorithm, CoverSumm, obtains significantly better runtime compared to baseline approaches. This is because centroid-based algorithms are generally much faster, brute force algorithm has a time complexity $O(n^2)$ with knn search being $O(n)$ per step. While the other extractive summarization paradigms utilize either the PageRank algorithm ($O(n^3)$ complexity) or eigen-decomposition ($O(n^4)$ complexity). Note that although we refer to extractive summarization systems by their approach (e.g., centroid-based), these algorithms are being used in state-of-the-art systems, like SemAE (Chowdhury et al., 2022) (e.g., Brute Force in Table 1 on SPACE is identical to SemAE).

**Automatic Evaluation of Summary Quality**. In this experiment, we probe the quality of the generated online summaries. Since we do not have access to human-written summaries at every time step, we construct silver extractive summaries by greedily selecting sentences (seen till time step $t$) with high ROUGE overlap with final human-written ones. We perform this experiment on the SPACE (Angelidis et al., 2021) dataset and report the ROUGE overlap with the silver summaries. We measure different ROUGE scores after every 20 steps and report the average score.

---

[1]Exact methods on Amazon US reviews achieve slightly less than 100% due to numerical stability issues in the cover tree implementation in edge cases, which can handle up to 32 bit numbers.

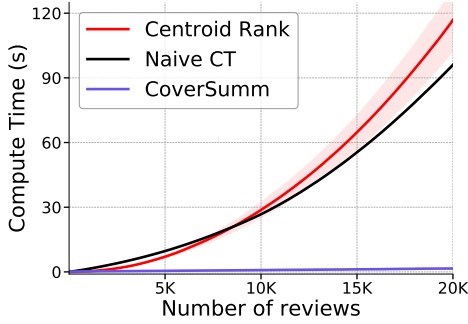

Figure 3: Time required by `CoverSumm` compared to baseline algorithms with an increasing number of reviews. We observe that the processing time of brute-force CentroidRank and Naive CT gradually increases, while the processing time of `CoverSumm` only slightly increases during summarization.

| Algorithm | $\overline{\text{R1}}$ | $\overline{\text{R2}}$ | $\overline{\text{RL}}$ | Time (s) |
|---|---|---|---|---|
| LSA | 25.04 | 4.66 | 15.34 | 825.57 |
| CentroidOPT | 27.74 | 4.82 | 16.39 | 263.84 |
| TextRank | 30.84 | 5.27 | 17.40 | 1147.89 |
| LexRank | 30.04 | 5.70 | 17.59 | 160.63 |
| SumBasic | 31.55 | 5.00 | 15.66 | 287.80 |
| Naive HNSW | 43.53 | 14.61 | 26.17 | 3.25 |
| Naive Faiss | 43.48 | 14.51 | 25.99 | 1.30 |
| `CoverSumm` (random) | 39.19 | 11.64 | 22.92 | 1.02 |
| `CoverSumm` (decay $\lambda$) | 41.03 | 12.18 | 24.03 | 1.35 |
| `CoverSumm` | 42.46 | 14.06 | 25.15 | 1.14 |

Figure 4: Average ROUGE scores obtained by different incremental summarization systems on SPACE dataset. $\overline{\text{R1}}$, $\overline{\text{R2}}$, $\overline{\text{RL}}$ denote the average ROUGE-1, ROUGE-2, and ROUGE-L scores respectively. We also report the time taken for incremental summarization per entity by different algorithms.

In Table 4, we observe that `CoverSumm` achieves the best tradeoff between summarization performance (in terms of ROUGE scores) and runtime in the incremental setting. Here, we would like to emphasize that the focus behind designing `CoverSumm` is on improving the efficiency of centroid-based summarization systems, not on enhancing summary quality. Apart from ROUGE scores, we further evaluate the quality of the generated summaries by their ability to track content in user reviews and human evaluation. We report these experiments in the following section.

## 4.2 Analysis

In this section, we analyze the functioning of `CoverSumm` through various experiments. First, we gauge the quality of `CoverSumm`'s summaries using Amazon US reviews (He & McAuley, 2016), which contains real-world review sets. Since gold summaries are infeasible for such large review sets, we evaluate if the summaries mimic proxy measures like sentiment polarity and user ratings of the aggregate reviews.

**User ratings.** In Figure 5(a), we report the overall user rating (in green) and the summary rating (in blue), where the reviews arrive in a temporal order provided by the dataset. We observe that the summary ratings mimic the overall trends in the user ratings. The absolute difference between user and summary ratings was 0.42. In Figure 5(b) & (c), we simulate a drift in the user reviews by ordering them according to ratings from low to high or vice-versa. We observe that `CoverSumm` is able to tackle such scenarios where the summary ratings still track the aggregate ratings.

**Sentiment Polarity**. In this experiment, we probe the sentiment polarity of the generated summaries and verify if they are consistent with the sentiment of the aggregate reviews. We use the VADER (Hutto & Gilbert, 2014) to extract sentiment polarity. We assign summaries a sentiment polarity score based on the average polarity score of their individual sentences. In Figure 6(a) & (b), we report the summary polarity (in green), the average review polarity (in blue), and human-written summary's polarity (in black). We report the scores for two distinct entities in the SPACE dataset. We observe that the generated summaries' sentiment polarity generally follows trends in the overall review polarity while achieving a polarity score close to the human-written summary with an increase in reviews. In general, the summary's polarity doesn't exactly track the aggregate review polarity as many reviews are neutral, and selecting them would result in less informative summaries (more details in Appendix E.3).

**Aspect discovery.** In this experiment, we assess if `CoverSumm` can capture the underlying aspects in user reviews. To investigate this, we progressively fed the system reviews from each aspect (e.g., hotel reviews about food first, followed by service, and so on). In Figure 6(c), we compute the number of unique aspects in the summaries and reviews at different time steps. We found that summaries successfully captured new aspects as they were introduced in the review stream. Additionally, we calculated the average absolute

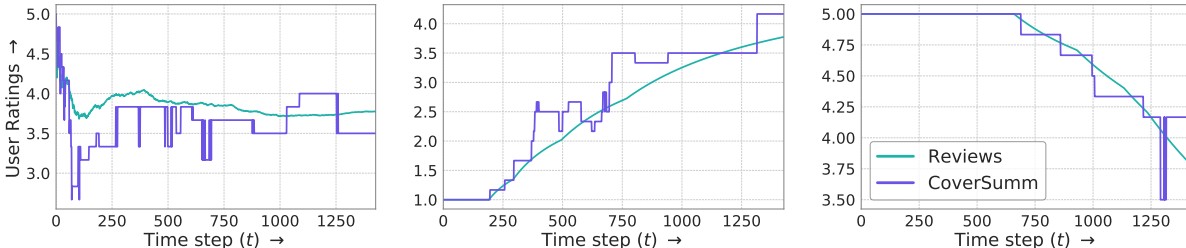

Figure 5: Evolution of user ratings in `CoverSumm`'s summary and user reviews during summarization in an incremental setting. The goal of this experiment is to determine if the user ratings can be accurately reflected in the incremental summaries from `CoverSumm`. We report the results in three settings when reviews arrive in their: (a) original temporal order; (b) ascending order of their ratings; (c) descending order of their user ratings. We observe that `CoverSumm`'s summary can track drifts in the ratings.

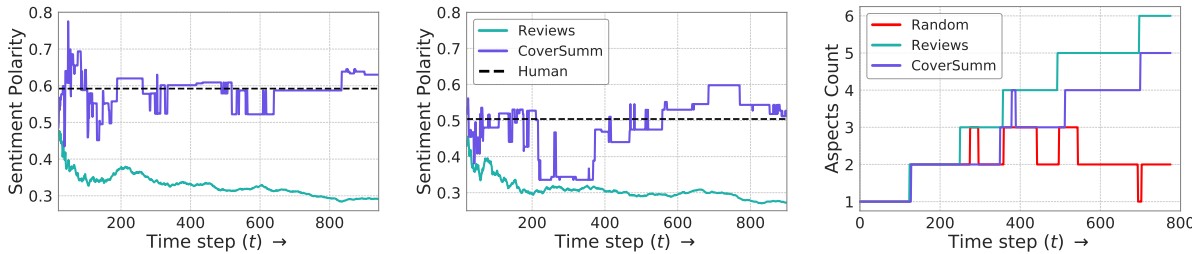

Figure 6: (a) & (b) Evolution of sentiment polarity in `CoverSumm`'s summaries in an incremental setting (for 2 entities in SPACE dataset), and (c) the number of unique aspects found in the `CoverSumm`'s summary and aggregate reviews. Random denotes the `CoverSumm` (random) baseline. These experiments show that the generated summaries can track finer aspects of the user reviews like sentiment polarity and aspects.

difference between the number of unique aspects in the reviews and summaries to be 0.32. This demonstrates that `CoverSumm` effectively captures the underlying semantics and selects relevant aspects from reviews.

**Number of reservoir search queries** ($n_{\mathrm{rs}}$). In this experiment, we probe the number of reservoir search queries (the most expensive step in Algorithm 1) executed by `CoverSumm`. In Figure 7 (a), we visualize the `rs` queries using data sampled from a uniform distribution and observe that `CoverSumm` performs a small number of queries (e.g., less than 150 queries when the number of points is 10K). We also observe that with an increasing number of points, the gap between subsequent reservoir search queries increases. This is expected as the nearest neighbours of the centroid change less frequently. We repeat the same experiment with different data dimensions and observe an increase in reservoir search queries with increasing data dimension, $D$. This is expected as the confidence for the bound in Proposition 2 is $(1-\delta)^{2D}$ decreases for higher data dimension, $D$. Therefore, nearest neighbour candidates can be outside the reservoir more frequently.

**Reservoir size** ($|\mathcal{R}|$). In this experiment, we investigate the size of $\mathcal{R}$ during summarization. We perform a synthetic experiment where points are randomly sampled from a uniform distribution. In Figure 7(b), we observe that the reservoir size at any time step is significantly small ($< 100$) compared to the number of points ($\sim$10K). We also experiment by varying the summary budget, $k$, and observe a nearly linear increase in reservoir size as predicted by Proposition 5.

**Data distribution ablations.** In this experiment, we investigate `CoverSumm`'s performance (time) when the input samples arise from a multi-modal distribution. Specifically, we sample $x \sim \sum_{i=1}^{m} a_i \mathcal{N}(\mu_i, \mathbf{I})$ from multi-dimensional gaussian distributions, where $a_i = 1/m$, $\mu_i = i\mathbf{1}$ and $m$ is the number of modes. In Figure 7(c), we observe that the time required by `CoverSumm` gradually increases with the number of modes in the input distribution. This behavior is expected as the number of representations near the centroid may change frequently as the number of modes increases.

**Human Evaluation**. To assess the quality of the summaries generated by `CoverSumm`, we perform a human evaluation of the generated summaries from SPACE and Amazon US reviews dataset. We compare `CoverSumm`

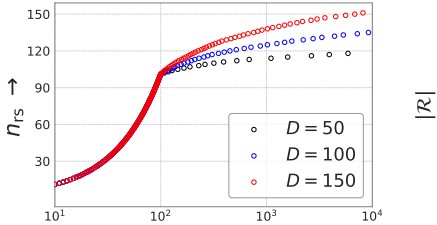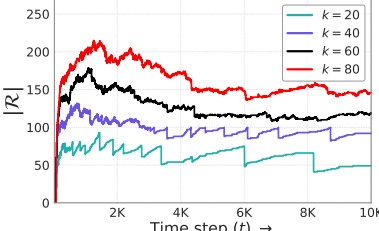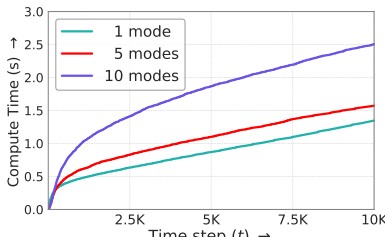

Figure 7: Plots showcasing (a) the cumulative number of reservoir search queries ($n_{rs}$), (b) the variation in reservoir size $|\mathcal{R}|$, and (c) compute time when points are sampled from a multi-modal distribution.

with its two variants: `CoverSumm` (random) and `CoverSumm` (decay $\lambda$). We perform human evaluation experiments on Amazon MTurk. Since evaluating the entire set of summaries is expensive, we selected consecutive summary pairs, which had at least one change in them. Specifically, given a summary pair, we ask the annotator to judge whether the change was: (a) *redundant*: if the information was already provided in the previous summary, and (b) *informative*: if the new selected review sentence is informative or "summary worthy" (e.g., a sentence like 'The boots wore off after a month' is summary worthy while a sentence like 'I do not like them' is not). We provide more details about the human evaluation setup in Appendix E.2.

In Table 2, we report the percentage of summary pairs marked as redundant or informative. On AMAZON, we observe that `CoverSumm` generates the least redundant summaries (only 8.8% are marked as redundant) while making the most informative changes (around 65% of the time). On SPACE, we observe that different systems achieve similar performance

| | AMAZON | | SPACE | |
|---|---|---|---|---|
| Algorithm | Redund. ↓ | Info. ↑ | Redund. ↓ | Info. ↑ |
| `CoverSumm` (random) | 28.2% | 54.0% | **21.6**% | 78.2% |
| `CoverSumm` (decay $\lambda$) | 12.3% | 63.4% | 25.8% | 79.9% |
| `CoverSumm` | **8.8**% | **65.4**% | 24.1% | **80.7**% |

Table 2: Human evaluation results evaluating the redundancy and informativeness of generated summaries from `CoverSumm` and its variants.

with `CoverSumm` achieving the best results in informativeness. To put these results in perspective, we need to consider some of the key differences between AMAZON and SPACE datasets. SPACE reviews have a similar structure with sentences like "The room was great", "The staff was helpful", etc. On the other hand, AMAZON US Reviews have a much more diverse set of reviews e.g., "Overall I'm very pleased with the purchase", "When I got it I was surprised how soft the lamb skin is, very nice". We see these properties manifest in the human evaluation results. In the SPACE dataset, we hypothesize that when reviews are altered with similar positive information, this is generally seen as informative, yielding high informative scores. `CoverSumm` (random) algorithm may be less sensitive to redundancy scores in this setup as it randomly chooses instances to be in the reservoir thereby encouraging diversity in the final summary. Overall, this experiment illustrates the need for evaluation on a diverse set of reviews to effectively gauge the system's performance.

## 5 Conclusion

In this paper, we proposed `CoverSumm`, an efficient algorithm to perform centroid-based extractive summarization in an incremental setup. `CoverSumm` leverages cover trees to perform nearest neighbour search efficiently, thereby obtaining up to 36x speed improvement over naive baselines. We perform extensive theoretical and empirical analysis to show that `CoverSumm` generates high-quality opinion summaries that accurately track the semantics in the review set. Detailed analysis shows that `CoverSumm`'s performance is dependent on the underlying data distribution, and its efficiency can suffer in adversarial scenarios. Most works use empirical evaluations on specific datasets to choose a summarization technique. In this work, we focus on centrality-based measures for extractive summarization. Nevertheless, determining the optimal summarization paradigm for different domains remains an open question. Future work could explore the use of efficient data structures like CoverSumm or improved confidence intervals (Waudby-Smith & Ramdas, 2024) to further improve efficiency/accuracy trade-offs across a broad class of summarization methods.

## 6 Acknowledgements

The authors are thankful to Michael Boratko, Anneliese Brei, Rahul Kidambi, Haoyuan Li, and Anvesh Rao Vijjini for helpful feedback and discussions. Somnath Basu Roy Chowdhury and Snigdha Chaturvedi were partly supported by the National Science Foundation under award DRL-2112635.

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

# A  Theoretical Proofs

**Contents**

## A.1  Hoeffding's Inequality

**Theorem 1** (Hoeffding-Azuma inequality (Hoeffding, 1963)). *Let $x_1, \ldots, x_t$ be i.i.d. random variables supported on $[-b/2, b/2]$. Define $\mu = \mathbb{E}[x_1]$ and $\hat{\mu} = \frac{1}{t} \sum_{i=1}^{t} x_i$. Then, for any $\delta \in (0, 1]$, with probability at least $(1 - \delta)$, it holds that*

$$|\mu - \hat{\mu}| < \sqrt{\frac{b^2 \log(2/\delta)}{2t}} \tag{6}$$

## A.2  Proof of Proposition 1

*Proof.* We proceed by applying the Hoeffding inequality (Theorem 1) to individual dimensions of the estimated centroid vector as shown below:

$$\mu_t^{(i)} = \mu^{(i)} \pm \sqrt{\frac{b^2 \log(2/\delta)}{2t}}.$$

Using triangle inequality, we can bound the Euclidean distance to $\mu_t$ from any point $x \in \mathbb{R}^D$:

$$\|\mu_t - x\| = \|\mu - x + \epsilon\|, \text{ where } \epsilon^{(i)} = \pm\sqrt{\frac{b^2 \log(2/\delta)}{2t}}$$

$$\leq \|\mu - x\| + \|\epsilon\|$$

$$= \|\mu - x\| + \sqrt{\frac{Db^2 \log(2/\delta)}{2t}}.$$

Therefore, for an Euclidean distance metric $d(\cdot, \cdot)$, we have obtained the following result:

$$d(\mu_t, x) = d(\mu, x) + \sqrt{\frac{Db^2 \log(2/\delta)}{2t}}.$$

$\square$

The above proof can be extended to any distance metric that follows triangle inequality. The new bound can be written as: $d(\mu_t, x) \leq d(\mu, x) + \|\epsilon\|$, where $\epsilon = \left[\pm\sqrt{b^2 \log(2/\delta)/2t}\right]^D$.

## A.3  Proof of Proposition 2

*Proof.* We estimate the distance between subsequent centroids using the triangle inequality of Euclidean distance and use the results from Proposition 1 as follows:

$$d(\mu_t, \mu_{t+i}) \leq d(\mu_t, \mu) + d(\mu_{t+i}, \mu)$$

$$\leq d(\mu, \mu) + \sqrt{\frac{Db^2 \log(2/\delta)}{2t}} + d(\mu, \mu) + \sqrt{\frac{Db^2 \log(2/\delta)}{2(t+i)}}$$

$$\leq 2\sqrt{\frac{Db^2 \log(2/\delta)}{2t}}$$

$$= \sqrt{2Db^2 \log(2/\delta)/t}.$$

$\square$

### A.4 Proof of Proposition 4

*Proof.* To prove this proposition, we first note that $\mathcal{R}$ is initialized using reservoir search with radius $r = d_k + \lambda$, where $d_k$ is the distance of $\mu_t$ to its $k$-th neighbour. Therefore, for any radius $r \geq d_k$, the reservoir size is always $|\mathcal{R}| \geq k$.

Next, we consider the case when the updated centroid $\mu_t$ is within distance $\|\mu_t - \mu_{\text{last}}\| < \lambda/2$. In this case, the distance from $\mu_t$ of any representation not in the reservoir (blue circles ● in Figure 2 (left & center)) is at least $d(x, \mu_t) > d_{\min} = d_k + \lambda/2$, $\forall x \in \mathcal{X}_t \setminus \mathcal{R}$. Now, note that all points in $\mathbf{S}_{\text{last}} \in \mathcal{R}$, as the $\mathcal{R}$ has not been updated after that (also $|\mathbf{S}_{\text{last}}| = k$). Next, we can show that:

$$\forall x \in \mathbf{S}_{\text{last}}, \ d(x, \mu_t) \leq d(x, \mu_{\text{last}}) + d(\mu_t, \mu_{\text{last}}) \tag{7}$$
$$< d_k + \lambda/2 = d_{\min}$$

This shows that the reservoir has at least $k$ points within $d_{\min}$ of $\mu_t$. Therefore, performing knn within $\mathcal{R}$ is exact. When the updated mean $\mu_t$ drifts more $\Delta \geq \lambda/2$ (shown in Figure 2 (right)), we perform a reservoir search query on the entire cover tree which is also exact. $\square$

### A.5 Proof of Proposition 5

*Proof.* First, we note that reservoir search queries on the cover tree are executed when the bound obtained in Proposition 2 is violated. This happens with probability $p_t = 1 - (1-\delta)^{2D}$ at each time step $t$. Therefore, the expected value of $n_{\text{rs}}$ can be obtained as follows:

$$n_{\text{rs}} = \sum_t p_t$$

$$= \sum_t 1 - (1-\delta)^{2D}$$

$$= \sum_t 2D\delta - D(2D-1)\delta^2 + \dots$$

$$\leq \int_{t=1}^{n} \left[ \frac{2D}{t} + \frac{D(2D-1)(2D-2)}{3t^3} + \dots \right] dt$$

$$\approx \int_{t=1}^{n} \left[ \frac{2D}{t} + \frac{D(2D-1)(2D-2)}{3t^3} + \dots \right] dt$$

$$= 2D \log n + \frac{D(2D-1)(2D-2)}{6n^2} - \dots$$

$$= O(D \log n).$$

We set $\delta = \Theta(1/t)$ as used in our algorithm and assume that $n \gg D$ in the above proof. $\square$

### A.6 Proof of Proposition 6

*Proof.* For a metric space $(\mathcal{X}, d)$, we assume the doubling constant to be $c$. The size of the reservoir $|\mathcal{R}|$ is given by the maximum number of elements in the ball $B(\mu_n, d_k(1 + \epsilon))$, where $\mu_n$ is the last query for range

search in the cover tree, and $d_k(1 + \epsilon)$ is the radius used for the same. We know that $|B(\mu_n, d_k)| = k$. Using this, we proceed to formulate the bound as follows:

$$
\begin{aligned}
|\mathcal{R}| &\leq |B(\mu_n, d_k(1 + \epsilon))| \\
&\leq c^{\lceil \log_2(1 + \lambda/d_k) \rceil} |B(\mu_n, d_k)| \\
&\leq c^{\lceil \log_2(1 + \sqrt{\alpha D \log(2t)/2t}/d_k) \rceil} k.
\end{aligned}
\tag{8}
$$

From Equation 8, we observe that we can control the reservoir size using the hyperparameter $\alpha$. Next, we show that the reservoir size bound is only dependent on data dimension $D$, number of nearest neighbour $k$, and aspect ratio $\Delta$ of $X$. For large $t \gg 0$, we have the limit: $\lim_{t \to \infty} \frac{\log(2t)}{2t} \to 0$. Plugging this result in Equation 8, we get the following

$$
|\mathcal{R}| \leq c^{\lceil \log_2(1 + \epsilon) \rceil} k = ck,
\tag{9}
$$

where for large $t \gg 0$, $\epsilon \to 0$ is a small constant. $\qquad\square$

**Discussion**. Therefore, at later time steps, we observe that $|\mathcal{R}|$ is directly proportional to the number of nearest neighbours, $k$. For a metric space $(X, d)$, the doubling constant $(c)$ can be bounded as $c \leq 2^{D\lceil 1 + \log_2 \Delta \rceil}$, where $\Delta$ is the aspect ratio of the space. The aspect ratio $\Delta$ is defined as the ratio of the largest to the smallest interpoint distance in that space.

## A.7 Additional Theoretical Results

**Proposition 7** (Interval between reservoir search queries). *The number of steps (interval) between two consecutive reservoir search queries $(m)$ grows linearly with the time step, i.e., $m = O(t)$.*

*Proof.* The evolving centroid is computed in $O(1)$ time as shown below:

$$
\mu_{t+1} = \left(1 - \frac{1}{t+1}\right)\mu_t + \frac{1}{t+1}\mathbf{x}_{t+1}.
$$

In a similar manner, we can compute $\mu_{t+2}$ as follows:

$$
\begin{aligned}
\mu_{t+2} &= \left(1 - \frac{1}{t+2}\right)\mu_{t+1} + \frac{1}{t+2}\mathbf{x}_{t+2} \\
&= \left(1 - \frac{1}{t+2}\right)\left(1 - \frac{1}{t+1}\right)\mu_t \\
&\quad + \left(1 - \frac{1}{t+2}\right)\frac{1}{t+1}\mathbf{x}_{t+1} + \frac{1}{t+2}\mathbf{x}_{t+2} \\
&= \frac{t}{t+2}\mu_t + \frac{\mathbf{x}_{t+1} + \mathbf{x}_{t+2}}{t+2}.
\end{aligned}
$$

Inductively, we have the following:

$$
\mu_{t+m} = \frac{t}{t+m}\mu_t + \frac{\sum_{i=1}^m \mathbf{x}_{t+i}}{t+m}.
$$

Now let us consider that the last reservoir search query was executed at time step $t$ $(\mu_{\text{last}} = \mu_t)$. We compute the drift from $\mu_{\text{last}}$ after $m$ steps as follows:

$$
\begin{aligned}
\Delta &= \|\mu_{\text{last}} - \mu_{t+m}\| \\
&= \left\|\left(1 - \frac{t}{t+m}\right)\mu_{\text{last}} - \frac{\sum_{i=1}^m \mathbf{x}_{t+i}}{t+m}\right\| \\
&= \frac{m}{t+m}\left\|\mu_{\text{last}} - \sum_{i=1}^m \mathbf{x}_{t+i}/m\right\| \\
&= \frac{m}{t+m}\|\mu_{\text{last}} - \mu_{t+1:t+m}\|,
\end{aligned}
$$

where $\mu_{t+1:t+m}$ is the mean of $m$ representations observed after the last reservoir search query. Now, let us consider the scenario where the centroid drift just exceeds the threshold, $\Delta \approx \frac{\lambda}{2}$:

$$\frac{m}{t+m}\|\mu_{\text{last}} - \mu_{t+1:t+m}\| \approx \frac{\lambda}{2}$$
$$m \approx \frac{\lambda t}{2\|\mu_{\text{last}} - \mu_{t+1:t+m}\| - \lambda}$$
$$m \sim O(t),$$

where $\lambda$ is confidence bound defined in Equation 4. □

**Discussion**. We observe that the interval between consecutive reservoir search queries $m$ is directly proportional to the time step of the last query $t$ ($\lambda$ remains constant for any reasonably large $t$). This shows that the reservoir search queries become infrequent as more samples are processed. We also observe that $m$ reduces as the $\|\mu_{\text{last}} - \mu_{t+1:t+m}\|$ increases, which implies that the reservoir search queries become more frequent whenever there is a shift in the underlying input distribution.

**Proposition 8** (Interval between reservoir search during distribution drift). *The number of steps (interval) between two consecutive reservoir search queries (m) when the centroid of the distribution drifts, increases with the number of representations processed $m = O(\sqrt{t \log t})$.*

*Proof.* Let us consider the scenario where the centroid of the distribution from which representations are sampled shifts from $\mu_1$ to $\mu_2$ (where $\mu_1, \mu_2 \in \mathbb{R}^d$). The unit vector in the direction of the drift is given as $\bar{v} = \frac{\mu_1 - \mu_2}{\|\mu_1 - \mu_2\|}$. During the drift, we assume that representations are sampled from the dynamic distribution $\mathbf{x} \sim \mathcal{N}(\mu_t + \beta\bar{v}, \epsilon I)$, where $\epsilon \ll \beta$ and $\mu_t$ is the current mean of all samples seen so far. We assume a specific form of distribution shift as shown below:

$$\mu_{t+1} = \left(1 - \frac{1}{t+1}\right)\mu_t + \frac{x_t}{t+1}$$
$$= \left(1 - \frac{1}{t+1}\right)\mu_t + \frac{\mu_t + \beta\bar{v} \pm \epsilon\mathbf{1}}{t+1}$$
$$= \mu_t + \frac{\beta\bar{v} \pm \epsilon\mathbf{1}}{t+1}.$$

Extending this for $m$ steps, we get the following:

$$\mu_{t+m} = \mu_t + (\beta\bar{v} \pm \epsilon\mathbf{1})\sum_{i=t}^{t+m}\frac{1}{i+1}$$
$$\mu_{t+m} - \mu_t = (\beta\bar{v} \pm \epsilon\mathbf{1})\left(\sum_{i=1}^{t+m}\frac{1}{i} - \sum_{i=1}^{t}\frac{1}{i}\right)$$
$$\|\mu_{t+m} - \mu_t\| \leq \beta\left(\sum_{i=1}^{t+m}\frac{1}{i} - \sum_{i=1}^{t}\frac{1}{i}\right).$$

Incorporating the upper bound on the harmonic series, $\sum_{k=1}^{m}\frac{1}{k} = \log(m) + \epsilon_m$, with $\epsilon_m \geq 0$, and monotonically decreasing.

$$\|\mu_{t+m} - \mu_t\| \leq \beta\left(\log(t+m) - \log(t)\right)$$
$$= \beta\log\left(1 + \frac{m}{t}\right).$$

Now assuming $\mu_{\text{last}} = \mu_t$ is the last time the reservoir search query was executed. We compute the number of steps till the centroid drift $\Delta \geq \frac{\lambda}{2}$ exceeds the threshold:

$$\beta \log(1 + \frac{m}{t}) \geq \frac{\lambda}{2}$$
$$m \geq t\left(\exp(\lambda/2\beta) - 1\right)$$
$$= t\left(\exp\left(\frac{1}{2\beta}\sqrt{\frac{\alpha D \log(2t)}{2t}}\right) - 1\right)$$
$$\approx t\left(1 + \frac{1}{2\beta}\sqrt{\frac{\alpha D \log(2t)}{2t}} - 1\right)$$
$$= \frac{1}{2\beta}\sqrt{\frac{\alpha D t \log(2t)}{2}}$$
$$\sim O\left(\sqrt{t \log t}\right).$$

In this derivation, for large $t \gg 0$, the term $\lambda/2\beta$ is quite small allowing us to write the exponent term as $e^x = 1 + x$, which provides a final complexity of $O(\sqrt{t \log t})$. $\qquad\square$

**Discussion**. The above result shows that the interval between consecutive reservoir search queries is smaller when there is a distribution drift. We also observe that $m$ is inversely proportional to the amount of drift $\beta$, which implies that reservoir search queries are called more frequently when the amount of drift is large.

# B  Additional Related Work

**Extractive Opinion Summarization**. The task of extractive opinion summarization has been extensively researched by a long line of work (Kim et al., 2011; Zhong et al., 2019; Zhao & Chaturvedi, 2020; Zhong et al., 2020; Gu et al., 2022; Hosking et al., 2023; Mao et al., 2023; Siledar et al., 2023; Jiang et al., 2023; Sosea et al., 2023; Li & Chaturvedi, 2024). Most systems operate in an unsupervised setup, where saliency scores are assigned to review sentences and a subset of sentences are selected based on their saliency scores. Several forms of saliency computation methods have been explored by prior works using: raw textual frequency features (Nenkova & Vanderwende, 2005; Nenkova & Bagga, 2003), distance from centroid (Radev et al., 2004), graph-based approaches (Erkan & Radev, 2004; Mihalcea & Tarau, 2004), among others. More recent neural approaches (Angelidis et al., 2021; Chowdhury et al., 2022; 2023; Li et al., 2023) perform summarization in a two-step process by learning review representations followed by an inference algorithm to generate the summary using the learned representations. In our algorithm, we follow the centroid-based summarization paradigm and use a similar saliency scoring mechanism to perform opinion summarization incrementally.

**Timeline Summarization**. Our work on incremental opinion summarization is closely related to the task of timeline summarization. Timeline summarization, which is a form of temporal summarization similar to our setup, has been studied extensively for news articles (Allan et al., 2001; McCreadie et al., 2014; Ge et al., 2016; Ghalandari & Ifrim, 2020; Pratapa et al., 2023) and in multi-document settings (John & Asharaf, 2014; Manuvinakurike et al., 2021; Yoon et al., 2023; Laskar et al., 2023). This task involves generating a sequence of summaries that reflect the timeline of long-ranging news topics. Timeline summaries (Yan et al., 2011b;a; Steen & Markert, 2019; Chen et al., 2023) are expected to have the following features: summaries should discuss important events in a time segment and consecutive summaries should not be redundant. Recently, (Cheang et al., 2023) has shown several challenges involved in using language models for timeline summarization as pre-training of LMs affect the faithfulness of future summaries. In contrast to timeline summaries, opinion summaries in an incremental setup should capture the dominant opinions at each time step without any redundancy constraints, as only one summary is presented to the user at a time.

## C   Synthetic Data Generation

In this section, we describe the process of sampling synthetic representations from the LDA (Blei et al., 2003) process. The overall algorithm is presented in Algorithm 3. First, we sample the word distribution about each topic. Secondly, for every document or review, we estimate its length. Thirdly, we select the words within each review by initially sampling a topic, and subsequently, picking a word from that topic's specific word distribution. We consider 10 different topics (can be considered as aspects for opinion summarization), vocabulary size $|V| = 100$, average review length $\bar{L} = 150$, and generate a total of $n = 10^4$ reviews. We consider one-hot vectors for each word as the $\texttt{repr}(\cdot)$ function and retrieve the mean of all words as the review representation $e_i$. Finally, the entire set of synthetic representations $\mathcal{E}$ is returned to the user.

---
**Algorithm 3** Synthetic LDA data generation

---
1: **function** LDA(Topic Count $k$, Vocabulary Size $|V|$, Average review length $\bar{L}$, Reviews Count $n$).
2: $\quad$ $\alpha = \mathbf{1}^{1 \times k}$, $\beta = \mathbf{1}^{1 \times |V|}$
3: $\quad$ // sample word distribution per topic
4: $\quad$ $\phi = \mathrm{Dir}(\beta) \in \mathbb{R}^{k \times |V|}$
5: $\quad$ $\mathcal{E} = \{\}$ // empty representation set
6: $\quad$ **for** $i = 1 \dots n$ **do**
7: $\quad\quad$ // sample topic distr. and review length
8: $\quad\quad$ $\theta \sim \mathrm{Dir}(\alpha)$, $l_i \sim \mathrm{Pois}(\bar{L})$
9: $\quad\quad$ $\mathcal{W}_i = \{\}$ // words in the $i$-th review
10: $\quad\quad$ **for** $j = 1 \dots l_i$ **do**
11: $\quad\quad\quad$ // sample new word
12: $\quad\quad\quad$ $t \sim \mathrm{Multinomial}(\theta)$
13: $\quad\quad\quad$ $w \sim \mathrm{Multinomial}(\phi_t)$
14: $\quad\quad\quad$ $\mathcal{W}_i = \mathcal{W}_i \cup w$ // add the new word
15: $\quad\quad$ **end for**
16: $\quad\quad$ // mean word representation
17: $\quad\quad$ $e_i = \mathbb{E}_{w \sim \mathcal{W}_i}[\texttt{repr}(w)]$
18: $\quad\quad$ $\mathcal{E}_i = \mathcal{E}_i \cup e_i$ // add representation
19: $\quad$ **end for**
20: $\quad$ **return** $\mathcal{E}$
21: **end function**

---

## D   Review Deletion Using `CoverSumm`

In this section, we show that we can use `CoverSumm` to update summaries when a subset of reviews is deleted. This scenario may be useful when some reviews are flagged for their reliability or foul language. In this setting, the user provides the system with a set of sentences $\mathbf{x}_{\text{del}}$ to be deleted and expects the updated summary as the output. In Algorithm 4, we present the algorithm to retrieve the updated summary when a set of sentences are deleted. First, we remove the sentences from the overall sentence set $\mathcal{X}_t$, cover tree `ct`, and reservoir $\mathcal{R}$. Next, we observe that we need to perform reservoir search on the entire cover tree if either of the conditions is met: the reservoir size is less than $k$, or the new mean has drifted beyond $\lambda/2$ (Line 11). If neither of the above conditions are met, the algorithm can generate the summary from the existing reservoir $\mathcal{R}$. We observe that the most expensive step in Algorithm 4 is deletion from cover trees (Line 4), therefore executing the DELETE($\cdot$) function should at least take $O(m \log |\mathcal{X}_t|)$ time (where $m$ is the number of sentences to be deleted and $|\mathcal{X}_t|$ is the total number of elements in the cover tree during deletion).

---

**Algorithm 4** `CoverSumm` Deletion Routine

---

1: **function** DELETE(Sentences to delete $\mathbf{x}_{\text{del}}$, last centroid $\mu_{\text{last}}$, reservoir $\mathcal{R}$)
2:     $\mathcal{X}_t = \mathcal{X}_{t-1} \setminus \mathbf{x}_{\text{del}}$ // remove $\mathbf{x}_{\text{del}}$
3:     // delete $\mathbf{x}_{\text{del}}$ from the cover tree
4:     `ct.delete`($\mathbf{x}_{\text{del}}$)
5:     // computed incrementally in $O(|\mathbf{x}_{\text{del}}|)$
6:     $\mu_t = \mathbb{E}[\mathcal{X}_t]$
7:     // drift of $\mu_t$ from the last query $\mu_{\text{last}}$
8:     $\Delta = \|\mu_t - \mu_{\text{last}}\|$
9:     $\mathcal{R} = \mathcal{R} \setminus \mathbf{x}_{\text{del}}$ // delete $\mathbf{x}_{\text{del}}$ from $\mathcal{R}$
10:    // check if drift exceeds threshold or $\mathcal{R}$ has less than $k$ elements
11:    **if** $\Delta > \lambda/2 \ \vee \ |\mathcal{R}| < k$ **then**
12:        $\lambda = \sqrt{\frac{\alpha D \log(2/\delta)}{2(t-|\mathbf{x}_{\text{del}}|)}}$
13:        $\mathcal{R} = $ `ct.ReservoirSearch`($\mu_t, r, \lambda$)
14:        $\mu_{\text{last}} = \mu_t$
15:    **end if**
16:    $\mathbf{S}_t = $ `knn`($\mathcal{R}, \mu_t$)
17:    **return** $\mathbf{S}_t$
18: **end function**

---

# E Experiments

## E.1 Implementation Details

We implemented all our experiments in Python 3.6 on a Linux server. The experiments were run on a single Intel(R) Xeon(R) Silver 4214 CPU @ 2.20GHz processor. In our experiments, we set $\delta = 1/t$ and perform a grid search on a small held-out set to determine $\alpha$, for each dataset. The summary budget in all experiments was $k = 20$. In our experiments, we use the SG Tree implementation available in `graphgrove` library.[2] We will make the codebase, along with detailed instructions on how to replicate our experiments, publicly available after the double-blind review phase. We used the default set of hyperparameters available with HNSW and FAISS.

## E.2 Human Evaluation Details

We performed human evaluation experiments on Amazon MTurk. We considered entities from the Amazon US reviews dataset where the number of reviews was more than 1000. In Table 2, we report the percentage of summary pairs marked as redundant or informative. All the datasets used for the evaluation were in English language and we did not perform additional data collection on our own. Human judges were compensated at a wage rate of $15 per hour.

## E.3 Additional Experimental Results

**Adversarial distributions**. A limitation of `CoverSumm` is that its performance is dependent on the input data distribution. If the input points are not bounded, and the centroid of the input representations shifts frequently, `CoverSumm`'s performance worsens as we need to perform $k$NN queries on the cover tree quite often. To simulate this scenario, we sample points from a dynamic distribution where the mean shifts continuously. Specifically, we sample the $i$-th representation $x_i \sim \mathcal{N}(\mu_i, \Sigma)$, where $\mu_i = (i/n)\mathbf{1}$ and $n$ is a hyperparameter. In Table 3, we observe that `CoverSumm` is quite slow in this scenario and obtains similar time complexity to the brute force method. We

| Algorithm | Time (s) ↓ |
|---|---|
| Brute force | 25.51 |
| Naive CT | 14.75 |
| `CoverSumm` (reservoir) | 15.36 |
| `CoverSumm` (reservoir + lazy) | 15.39 |

Table 3: Runtime of different algorithms when the points are sampled from an adversarial distribution.

also observe that the Naive CT is the fastest algorithm in this setting, as it does not have additional range search and reservoir computations. However, adversarial distributions are rare in real-world settings. In such scenarios, we recommend using Naive CT instead of `CoverSumm`.

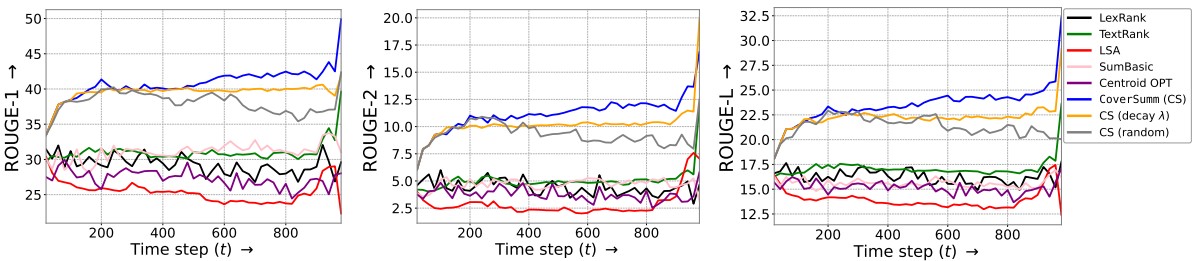

Figure 8: Evolution of ROUGE scores during incremental summarization. `CoverSumm` achieves the best scores with a gradual increase over time as more reviews are observed. `CoverSumm` is abbreviated as CS.

**Summarization Quality**. In this experiment, we investigate how the quality of the generated summaries improves in the incremental setup. Specifically, we measure the ROUGE scores of the generated summaries with the human-written summaries for all the reviews and report their evolution over time. In Figure 8, we observe a gradual improvement in summary quality (in terms of ROUGE overlap with the human-written

---

[2]https://github.com/nmonath/graphgrove/

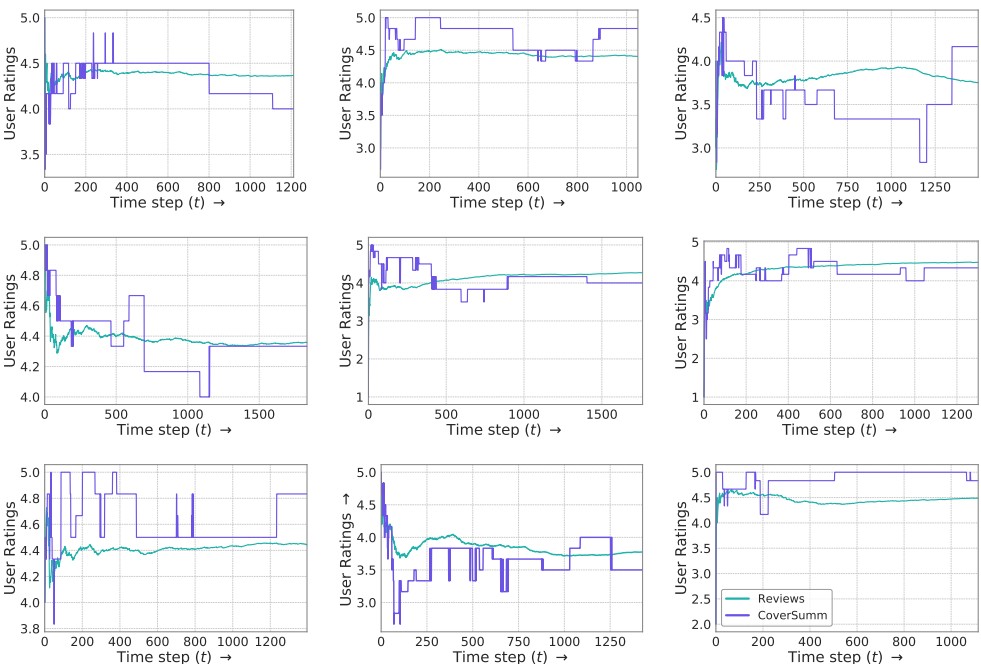

Figure 9: Evolution of average user ratings in the summary and user reviews for different products in the Amazon US reviews dataset. We observe that the user ratings in `CoverSumm`'s summaries can track the aggregate review ratings closely for all products.

summary) across different summarization systems. We also observe that `CoverSumm` and its variants achieve the best ROUGE scores throughout the incremental summarization process.

**User ratings**. We report the evolution of average user ratings and summary ratings using `CoverSumm` during the incremental summarization process. In Figure 9, we showcase the summary rating trajectory for 9 different product entities from the Amazon dataset. We observe that for all instances the average summary rating mimics the aggregate user ratings across all time steps during summarization.

**Generated Summaries**. In Table 4, we report the summaries generated by `CoverSumm` over time for an entity in the SPACE dataset. We observe that the extracted summaries can be uninformative or redundant when there are only a few reviews, but the quality of summaries improves as more reviews pour in over time.

**Sentiment Polarity**. We report the evolution of sentiment polarity of `CoverSumm`'s summaries and aggregate reviews in Figure 10 for 21 different products from the SPACE dataset. For all products, we observe that the summary is able to capture the trends of sentiment polarity change in the original reviews. The generated summary also achieves sentiment polarity scores that are close to the aggregate user reviews.

| Time Step | CoverSumm Summaries |
|---|---|
| $t = 10$ | I've been at all three of the Key West Marriott Hotels and this is my favorite for value and relaxation. Has a great relaxing pool layout, full continental+ breakfast and on this last visit a staff which worked harder than most travel agents finding some last minute activities and making the bookings for our family. I did complain about the room at reception and they apologised and we were only staying the one night. |
| $t = 50$ | We took a 2 night 3 day vacation and stayed at the Fairfield Inn in January 2012. The bathroom smelled like sewer water, the front door couldn't open wider, it felt like we were in a motel. The location was good and the price reasonable for Key West. The Fairfield Inn is the absolute best value and the staff is nothing short of fantastic. Rooms nothing special; not very spacious and "weird" bathroom. Good location. |
| $t = 100$ | The staff is so hospitable, and the hotel was very reasonably priced and a good location. We had a good time at the pool and the bar is great. We took a 2 night 3 day vacation and stayed at the Fairfield Inn in January 2012. The location was good and the price reasonable for Key West. The staff was very Friendly and helpful. The room was very clean. Good location. |
| $t = 300$ | The hotel staff was very friendly and accommodating. The front desk people were very helpful and friendly. Staff was friendly and helpful. We would definitely stay here again. Rooms were nice and clean. We would definitely stay there again and would highly recommend this hotel. The pool area is very nice. Breakfast was more than adequate and the pool area was very enjoyable. The room was very clean. They succeeded! This hotel is great. |
| $t = 500$ | The hotel staff was very friendly and accommodating. The front desk people were very helpful and friendly. Staff was friendly and helpful. Rooms were nice and clean. We would definitely stay here again. Staff are very courteous and professional. The pool area is very nice. Breakfast was more than adequate and the pool area was very enjoyable. We would definitely stay there again and would highly recommend this hotel. They succeeded! The rooms are loud. |
| $t = 915$ (final) | Staff was friendly and helpful. The front desk people were very helpful and friendly. The hotel staff was very friendly and accommodating. The pool was nice, but we did not use it. Rooms were nice and clean. We would definitely stay here again. The pool area was very nice. A big plus was the free parking and large selection for continental breakfast. Staff are very courteous and professional. We were not disappointed! They succeeded! |

Table 4: Generated summaries from CoverSumm at different time steps for an entity from the SPACE dataset.

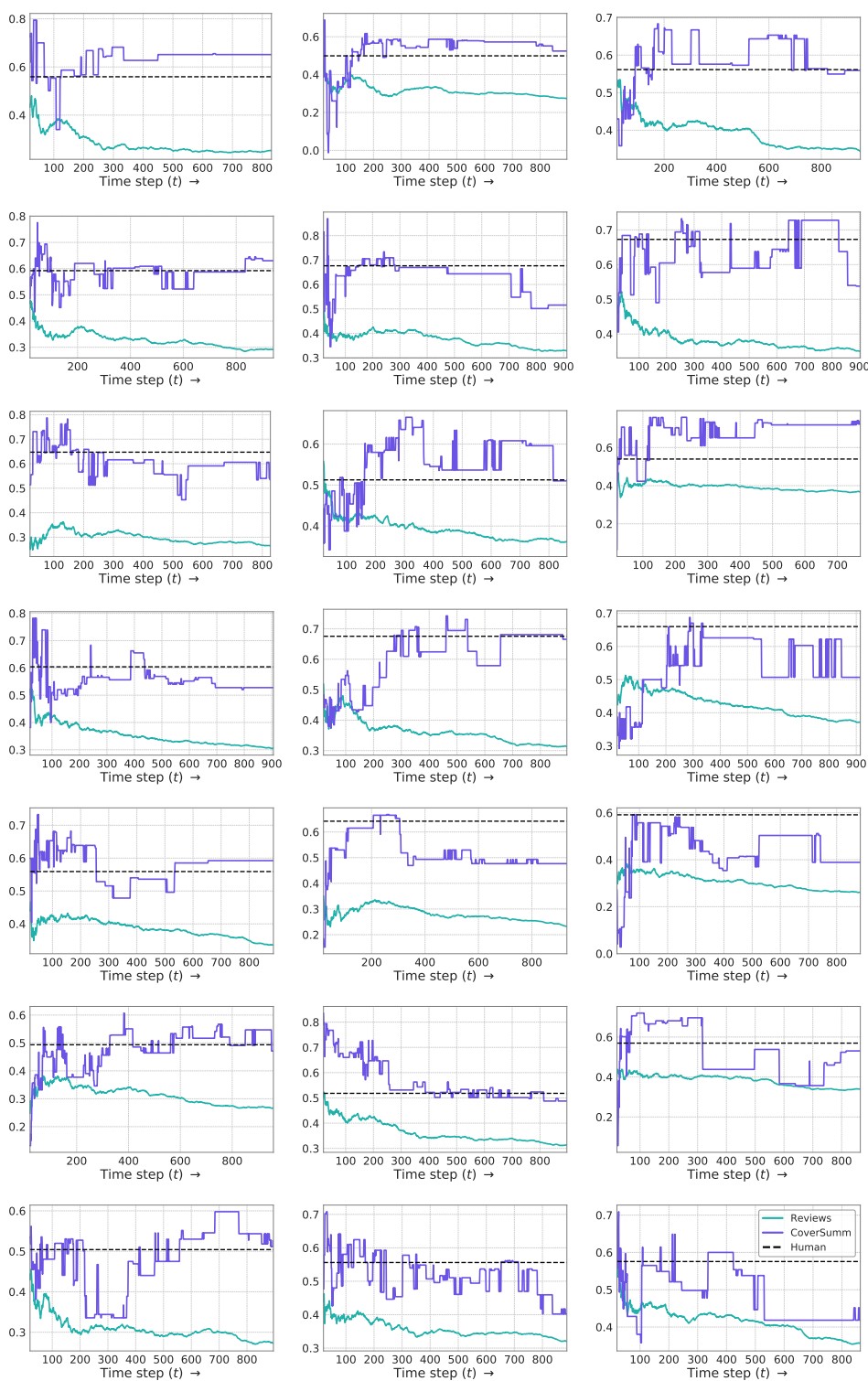

Figure 10: Evolution of sentiment polarity in `CoverSumm`'s summaries and aggregate user reviews for different products in the Amazon US reviews dataset. We observe that `CoverSumm`'s summaries can track the trends in sentiment polarity of the aggregate review set while achieving scores similar to the human-written summary.

