# OpenReview forum: "Incremental Extractive Opinion Summarization Using Cover Trees"
_TMLR — Accepted by TMLR_

### Review · Reviewer_Hkk2 · 2024-01-26

**Summary Of Contributions:**

A fast exact iterative k-nearest-neighbors to the centroid algorithm with theoretical analysis. Experiments showing how it performs for opinion summarization.

**Audience:**

Yes

**Claims And Evidence:**

Yes

**Requested Changes:**

- Please make sure the baselines (like CentroidOPT) are all clearly (if succinctly) described - critical
- Make clear how hyperparameters were set, and (especially) clarify how $\delta$ is chosen and what happens when you choose poorly - critical
- Add the R2's Naive Baseline to Table 1 and 2 - critical

**Strengths And Weaknesses:**

Thanks to the authors on the hard work on this paper.

Strengths
------------
- Their algorithm is general - it will work beyond the one specific use-case of iterative opinion summarization.
- Theoretical results (which I did not closely check) are nice to have
- Clear writing
- Good results (but I do ask for extra baselines below)

Weaknesses
----------------
- What's CentroidOPT? How come it's not as good as CoverSumm? In the intro you said "This objective, known as CentroidRank (Radev et al., 2004), functions by calculating the centroid of review sentences in the representation space, and selecting the nearest neighbors of the centroid as the summary". So it seems like CentroidOPT (which I assume is some sort of iterative version of CentroidRank) is trying to do the same thing as NaiveCT/CoversSumm.
- The actually reads like two different papers stapled together. Sections 2 and 3 are about a general-purpose algorithm where the goal is to keep a reservoir of neighbors to the centroid up to date as new data arrives. Section 4 is about an application of this general algorithm to one very specific setting with a few datasets. It's nice that you show CoverSumm works well in Table 2, but this is just performance on one dataset - is your intention to show that CoverSumm is a good general-purpose interative opinion summarization dataset? If so, you need more datasets with Rouge scores, probably in line with however-many are in your most recent competitor.
- What is the exact formula for $\delta$? How do you set it? Do you need cross-validation? Similarly, how were *all* the hyperparamters set? Both for your algorithm, and also everything else in Table 2?
- I don't understand the baseline CoverSumm (decay $\lambda$). What is the motivation here? At what rate do you decay? What is this baseline showing? What if you start with a bigger $\lambda$ than the normal one you set? You may want to get rid of this baseline...
- I think you need another baseline. Upon reading the description of the problem, my first instinct (I'll call this R2's Naive Baseline) was to just use a well-known approximate nearest neighbor package (HNSW or FAISS) + iterative computation of the centroid as follows:

(1) Initialize empty ANN and null centroid.
(2) For step $n$, given a new sample $x_n$, insert into ANN, and update centroid with $$\mu_n = \frac{1}{n} \left((n-1)\cdot\mu_{n-1}  + x_n\right)$$ (please double check my math)
(3) Issue the centroid as k-NN query to ANN.

I think this would be fairly easy to implement and run with off-the-shelf tools. There is the matter of setting hyperparameters for the ANN, but you can use whatever techniques you used for various algs in Table 2. Hopefully they were set in some rigorous way.

---

> ### Author Response · Authors · 2024-02-09
>
> Thank you for your detailed feedback and helpful comments. Please find our responses below.
>
> > Description of CentroidOPT
>
> CentroidOPT constructs the summary by greedily selecting sentences that maximize the generated summary’s similarity with the centroid. Specifically, at the $i$-th step a review sentence is selected using:
>
> $s_i = \arg\max_s \sim({S \cup s}, \mathrm{centroid})$, where $S$ is the current summary.
>
> This is different from CentroidRank, which selects $k$ most similar review sentences to the centroid. Due to this distinction, the summaries generated by CentroidOPT and CoverSumm are quite different.
> We have mentioned this in the current draft (Section Baselines, Page 8).
>
> > The proposed method is a bit general and not tied to a single application. Also, the overall goal of this work
>
> Regarding the concern about the work reading like two papers stapled together: In the introduction of our paper, we describe the opinion summarization task and its challenges, specifically in incremental settings. We then present an efficient algorithm to extend one of the state-of-the-art methods CentroidRank in the incremental setting. It is often common for a paper to describe a general-purpose method even when the experiments are focused on a single domain.
>
>
> The goal of our work is to present an efficient version of CentroidRank in the incremental summarization setting and not to develop a system that produces better-quality summaries than existing systems. CoverSumm produces the same summaries as CentroidRank, therefore its quality is the same as CentroidRank. Previous works [1, 2, 3] have focused on analyzing the quality of CentroidRank-based approaches.
>
> [1] Unsupervised Opinion Summarization Using Approximate Geodesics, Chowdhury et al., 2023, Findings of EMNLP.
>
> [2] Unsupervised Extractive Opinion Summarization Using Sparse Coding, Chowdhury et al., 2022, ACL.
>
> [3] Radev et al. 2004 Centroid-based summarization of multiple documents.
>
> > Formula for $\delta$ and how is it set?
>
> The exact formula is $\delta = \frac{c}{T}$, where $c$ is a hyperparameter set using grid search on a dev set and $T$ is the current time step during incremental summarization. All the hyperparameters were set using grid search on the dev set of the corresponding dataset.
>
> As mentioned, $\delta = \frac{c}{T}$, where $c$ is a hyperparameter set using grid search on a dev set and $T$ is the current time step during incremental summarization. We report the speed results when $c$ is varied on the synthetic dataset with uniform distribution.
>
> | c    | Time (s) |
> | -------- | ------- |
> | 100  | 0.91    |
> | 10  | 1.71    |
> | 1 | 2.10 |
> | 0.1    | 8.47   |
>
>
> We observe that the summarization time increases as we decrease $c$. This is expected as we increase $c$ (thereby $\delta$) the range query radius increases, thus more samples can be part of the reservoir. This would result in more NN queries to the CT if the reservoir is full. To counteract this, we can increase the reservoir size (set to 100 in the above table):
> For c=0.1, $c_{max} = 1000$, we get time: 1.24s.
>
>
> > Motivation behind CoverSumm (decay $\lambda$) baseline
>
> The motivation behind CoverSumm (decay \lambda) is to observe the effect of arbitrarily setting the threshold for reservoir search instead of using the result in Equation 4. For this baseline, we modify Line 9 in the CoverSumm algorithm to $\Delta \leq c_1 \exp^{-c_2T}$, where $c_1, c_2$  are hyperparameters. This result shows the utility of using the threshold we derived.
> We have added the details about this baseline in the main draft.
>
> > Approximate NN search baseline using HNSW and FAISS
>
> Thank you for the suggestion. The requested baseline is similar to the Naive CT results already reported in the paper. In Naive CT, the centroid is updated incrementally as you showed and we query the cover tree for NNs at each time step. There are different mechanisms to construct a graph index for efficient NN search: Cover trees, HNSW, and FAISS are a few options. We have now added the results for HNSW and FAISS in Tables 1 & 2.

---

> > ### Comment · Reviewer_Hkk2 · 2024-02-09
> >
> > Thank you for the updates.
> >
> > Having Naive FAISS in Tables 1 and 2 is very helpful.
> >
> > It seems like your algorithm is 4-5 faster than Naive FAISS (but with identical accuracy) on synthetic data, but is actually worse in R1/R2/R3 on the SPACE dataset, while being only negligibly faster.
> >
> > These results are troubling - they make the ultimate utility of this algorithm questionable. Can you show that it has a major speed increase on some real dataset without sacrificing ROUGE?

---

> > > ### Author Response · Authors · 2024-02-12
> > >
> > > Thank you for pointing this out.
> > >
> > > We found that the differences in the ROUGE scores come from differences in the ways ties are broken between nearest neighbour candidates (caused by numerical precision) in the implementation of FAISS and Cover Tree. We drew this conclusion by comparing the kNN outputs from FAISS and Cover Tree. We observed that in cases where the kNN differed, the distance between the distinct points was of the order $10^{-9}$.
> > >
> > >
> > > Ideally, we want to evaluate the performance of FAISS and Cover Tree, where such tie-breaking is not involved. To achieve this, we add a small amount of Gaussian noise, $\epsilon \sim \mathcal{N}(\mathbf{0}, 10^{-4})$, to break ties between representations. Using the same seed, we ensure that the same noise is added to both systems. We present results on 3 different seeds. We expect to observe the same performance.
> > >
> > >
> > > | Method    | ROUGE-1 | ROUGE-2 |ROUGE-L |
> > >  | ------- | -------- | ------- | -------- |
> > > |FAISS (Seed 1)|40.73| 12.79| 23.57 |
> > > |CoverSumm (Seed 1)|40.73| 12.79| 23.57 |
> > > |FAISS (Seed 2)|40.17| 11.60 |22.64 |
> > > |CoverSumm (Seed 2)|40.17| 11.60 |22.64 |
> > > |FAISS (Seed 3)|40.29| 12.61| 23.53|
> > > |CoverSumm (Seed 3)|40.29| 12.61| 23.53|
> > >
> > >
> > > From the above results, we observe that the performance of FAISS and CoverSumm are identical when tie-breaking is not needed. This shows that the small improvement in FAISS' performance resulted from its tie-breaking logic rather than fundamental algorithmic choices.
> > >
> > > Despite FAISS being a highly optimized implementation, our CoverSumm research implementation achieves ~13% speed improvement over FAISS in SPACE and 72% improvement in Amazon

---

> > > > ### Comment · Reviewer_Hkk2 · 2024-02-12
> > > >
> > > > Thanks, that is helpful.

---

### Review · Reviewer_LCuR · 2024-01-30

**Summary Of Contributions:**

This paper addresses a unique problem of incremental extractive summarization that is well suited in review summarization, where new product or service reviews arrive incrementally. While methods to extract summaries existed, such as direct predictions by deep learning models or centrality-based approaches, a method to do so efficiently in an incremental setting did not yet exist. The authors extended the centroid-based approach and proposed a new algorithm, CoverSum, that used a stable greedy (cover) tree and reservoir candidate representations to find the nearest neighbors and update the centroid efficiently.

The authors provided theoretical proof that bounded the efficiency (number of queries and reservoir size) of CoverSum. In the experiments, the authors used both synthetic and real-world datasets. The authors showed that CoverSum ran faster than other baseline algorithms, including brute force and naive cover tree, while returning extract nearest neighbors. In addition, the extracted summaries achieved high ROUGE scores and tracked the changes in user rating and sentiment. Additional analyses also supported the theoretical bounds.

**Audience:**

Yes

**Broader Impact Concerns:**

There is no ethical concern.

**Claims And Evidence:**

Yes

**Requested Changes:**

### Changes
1. I think the authors should include the performance of both real-world datasets in the main text. While many experiment results in the appendix seemed duplicates, some were conflicting or omitted. For example, Table 4 conflicts with Table 3 (with a good reason).
1. It might be possible to compute the ROUGE scores of the silver summaries.
1. Would it be possible to simulate the reviews "expiring" over time to demonstrate the deletion cases?


### Minor changes
1. Section 2.2 contained both $l$ and $\ell$. I believed they were meant to be the same symbol.
1. 25x was coming from the synthetic dataset of 100D vectors. I think the authors should make this claim more realistic.

**Strengths And Weaknesses:**

### Strengths

1. The paper addressed a unique aspect of the summarization problem that fits real-world applications.
2. The paper provided both theoretical bounds and empirical results. The reduction in runtime achieved was outstanding.
3.  The paper included additional analyses to confirm the theoretical bounds and highlight the strengths and weaknesses of the proposed algorithms.

### Weakness

1. Most of the paper was well written, but the evaluation section missed some important details.
    1. It is unclear how the "Time" in Table 1 was computed (i.e., total time of the whole simulation or average time per sentence added.)
    1. There were 10k reviews in the syntactic data, but how many entities?
1. Why was SemAE selected as a representation model? Was there any particular reason not to use general sentence embedding models? If there was one sentence to come in at a time, how would we obtain SemAE pre-trained with the datasets?
1. The omitted result might not fully support the claim. The Amazon review and the SPACE datasets were not similar as shown in Table 3 and Table 4. The inclusion of the Amazon review dataset in Table 1 should be necessary.
1. I appreciated that the author presented the human evaluation result, but the silver summaries' quality was still questionable.
1. No empirical evidence of the deletion case (i.e., reviews of a product evolved over time, getting better or worse.)

---

> ### Author Response · Authors · 2024-02-09
>
> Thank you for your detailed feedback and helpful comments. Please find our responses below.
>
> > Details about the (i) computational time and (ii) entity count for results in Table 1.
>
> (i) “Time” in Table 1 refers to the total time for the entire online summarization process per entity. Earlier, we had mentioned this in the caption of Table 1. Now, we have provided additional details in the Speedups section in the main text as well.
>
> (ii) We assume that the 10K review representations come from a single entity. We repeat this experiment 5 times and report the average results in Table 1. We have added these details in Section 4.1 (Main Results.)
>
> > Selection of SemAE for opinion summarization.
>
> Prior work [1] has shown that distributed representations from pre-trained models typically underperform compared to approaches such as SemAE. SemAE employs topical modeling within Transformer architecture yielding superior representations for extractive opinion summarization.
>
> Our approach, CoverSumm, assumes access to a representation learning model that works well for extractive opinion summarization tasks. SemAE was pre-trained in a self-supervised manner on user reviews from the training set of the SPACE dataset [2]. We conducted experiments using the test set from the SPACE dataset, which SemAE did not use during training.
>
> [1] Unsupervised Opinion Summarization Using Approximate Geodesics, Chowdhury et al., 2023, Findings of EMNLP.
>
> [2] Extractive Opinion Summarization in Quantized Transformer Spaces, Angelidis et al., 2020, TACL.
>
> > Speed results for Amazon dataset in Table 1.
>
> We have added the results on the Amazon dataset in Table 1. The additional results support the original claims.
>
> > Usefulness of silver summaries for evaluation
>
> Since we do not have a dataset with human-written incremental summaries, we felt extracted silver summaries are the best alternative. We have also reported how the ROUGE scores compared with the final human-written summary vary during the incremental summarization process (Figure 9 in Appendix E.3).
>
> We have also considered a setup where we iteratively summarize multiple entities and report the ROUGE scores at the boundary between two entities where we have access to the exact human written summaries. We observed similar trends in that setup too.
>
> The primary focus of our work is on enhancing the efficiency of CentroidRank within incremental summarization, rather than introducing a system aimed at generating higher-quality summaries. CoverSumm produces the same summaries as CentroidRank, therefore its quality is the same as CentroidRank. Previous works [1, 2, 3] have focused on analyzing the quality of CentroidRank-based approaches.
>
> [1] Unsupervised Opinion Summarization Using Approximate Geodesics, Chowdhury et al., 2023, Findings of EMNLP.
>
> [2] Unsupervised Extractive Opinion Summarization Using Sparse Coding, Chowdhury et al., 2022, ACL.
>
> [3] Radev et al. 2004 Centroid-based summarization of multiple documents.
>
> > Empirical evidence for review deletion scenario
>
> In our current draft, we have only mentioned that review deletion requests can be supported using CoverSumm. However, we haven't highlighted it as a primary contribution, and due to the lack of labeled datasets to properly evaluate such scenarios, we leave it up to future works to explore the deletion of user reviews using CoverSumm.
>
> > Human evaluation results for both real-world datasets
>
> Thank you for the suggestion. The slight variations in trends observed in the human evaluation results for SPACE and Amazon datasets stem from the distinct sets of reviews present in each dataset. We have incorporated the results of SPACE and Amazon in the main draft in Table 3 and added a discussion about the datasets as well.
>
> > ROUGE scores for silver summaries
>
> It is not possible to compute the ROUGE scores of the silver summaries as we do not have access to human-written summaries at every time step. We can calculate the ROUGE scores for the incremental silver summaries in comparison to the final human-written summary, but this evaluation may not accurately reflect their quality. This is because the initial summaries tend to achieve lower scores than the later ones.
>
> > Symbol change in Section 2.2
>
> We have fixed this in the current draft.
>
> > Modify the speed-up claims
>
> Thank you for bringing this to our attention. With the incorporation of the speed results for the Amazon dataset, we observe that we achieve up to 36x gains (because Amazon has more reviews). We have updated our claims accordingly in the paper.

---

### Review · Reviewer_izSt · 2024-02-01

**Summary Of Contributions:**

This paper presents an algorithm and theory for retrieving k nearest neighbors to a centroid when the data points arrive incrementally, and applies the algorithm to the summarization task. The algorithm maintains a small number of reservoir representations R that have a high chance of being selected as kNNs in the future. This reduces the execution time of kNN, focusing on the limited number of entries (R) instead of the entire entries. The main contribution of this work is the theoretical proofs that give the foundations for the maintenance of the reservoir. The experimental results confirm that the proposed algorithm finds the exact solution of kNNs in a shorter running time. The experiments also demonstrate the superiority of the proposed method on SPACE dataset over the baseline methods in terms of ROUGE scores and running time. This paper also reports various analyses, for example, user rating, sentiment polarity, and human evaluation.

**Audience:**

Yes

**Claims And Evidence:**

Yes

**Requested Changes:**

Minor point:

+ Change "Beygelzimer et al. (2006); Zaheer et al. (2019)" to "(Beygelzimer et al., 2006; Zaheer et al., 2019)" in P3.

**Strengths And Weaknesses:**

Strengths:

+ This paper is very well written with theoretical background and proofs.
+ The experiments show strong results and support the main claim of this paper.

Weaknesses:

+ The core of the proposal is not limited to update opinion summarization, but for k nearest neighbors with incremental data points. To be more precise, this paper does not include any proposal that is specific to opinion summarization. For example, the title of this paper can be something like "k nearest neighbors with incremental data points using cover trees." I'm wondering why the authors limit the scope of this work to opinion summarization.

---

> ### Author Response · Authors · 2024-02-09
>
> Thank you for your feedback and helpful comments. Please find our responses below.
>
> > Applications of the proposal algorithm beyond Opinion summarization
>
> Thank you for recognizing that our proposed approach of finding the k-nearest neighbours to the centroid is general and can be applied in different settings. Our focus on extractive summarization stems from the fact that CentroidRank-based approaches achieve state-of-the-art results and there is a practical need for having these systems function in an incremental setup. We considered applying our approach to select representative samples from various clusters in $k$-means clustering, but we struggled to identify an application where such a solution is used.
>
> > Request to update the citation style on Page 3.
>
> Thank you for pointing this out. We have fixed it in the updated draft.

---

### Decision · Action_Editor_itbF · 2024-03-11

**Recommendation:** Accept as is

**Comment:**

This paper presents an algorithm and theory for retrieving k nearest neighbors to a centroid when the data points arrive incrementally, and applies the algorithm to the extractive summarization task. After author rebuttal, it received Accept, Accept, Leaning Reject recommendations.

On one hand, reviewers agree that (1) the paper is well written with theoretical background and proofs; (2) experimental results support the main claim of this paper, the reduction of the proposed method in runtime is encouraging. So, overall, it is a nice executed paper.

On the other hand, two concerns remain. (1) After adding additional baselines during author rebuttal, it seem like the proposed method is only a bit faster than a naive baseline (Naive Faiss) and not much else. (2) The proposed method is general; this paper does not include any proposal that is specific to opinion summarization. It is a little bit pity that the authors limit the scope of this work to opinion summarization.

Overall, the merits of the paper outweigh its flaws; therefore, the Action Editor would like to recommend acceptance of the paper.

**Audience:**

Yes

**Claims And Evidence:**

Yes